# Distinct mesenchymal cell states mediate prostate cancer progression

Hubert Pakula[1,12], Mohamed Omar [1,2,12], Ryan Carelli[1,12], Filippo Pederzoli[1], Giuseppe Nicolò Fanelli [1,3], Tania Pannellini[1], Fabio Socciarelli[1], Lucie Van Emmenis [1], Silvia Rodrigues[1], Caroline Fidalgo-Ribeiro[1], Pier Vitale Nuzzo [1], Nicholas J. Brady [1], Wikum Dinalankara[1], Madhavi Jere [1], Itzel Valencia[1], Christopher Saladino[1], Jason Stone[1], Caitlin Unkenholz[1], Richard Garner[1], Mohammad K. Alexanderani[1], Francesca Khani [1], Francisca Nunes de Almeida[4], Cory Abate-Shen[4,5,6,7,8], Matthew B. Greenblatt [1], David S. Rickman [1], Christopher E. Barbieri [2,9], Brian D. Robinson [1,2,9], Luigi Marchionni [1] & Massimo Loda [1,2,10,11] ✉

In the complex tumor microenvironment (TME), mesenchymal cells are key players, yet their specific roles in prostate cancer (PCa) progression remain to be fully deciphered. This study employs single-cell RNA sequencing to delineate molecular changes in tumor stroma that influence PCa progression and metastasis. Analyzing mesenchymal cells from four genetically engineered mouse models (GEMMs) and correlating these findings with human tumors, we identify eight stromal cell populations with distinct transcriptional identities consistent across both species. Notably, stromal signatures in advanced mouse disease reflect those in human bone metastases, highlighting periostin's role in invasion and differentiation. From these insights, we derive a gene signature that predicts metastatic progression in localized disease beyond traditional Gleason scores. Our results illuminate the critical influence of stromal dynamics on PCa progression, suggesting new prognostic tools and therapeutic targets.

Prostate cancer (PCa) ranges from an indolent disease to aggressive, castration-resistant prostate cancer (CRPC), associated with a poor prognosis[1,2]. However, genetic alterations in epithelial cancer cells do not fully explain the different clinical behaviors of this malignancy[3,4].

Previous studies linked stromal gene expression to prostate carcinogenesis and progression[5–7], and our group described that stromal transcriptional programs vary in areas surrounding low vs. high Gleason score PCa. Notably, benign stroma is transcriptionally distinct in

[1]Department of Pathology and Laboratory Medicine, Weill Cornell Medicine, New York, NY 10021, USA. [2]Sandra and Edward Meyer Cancer Center, Weill Cornell Medicine, Belfer Research Building, 413 East 69th Street, New York, NY 10021, USA. [3]Department of Laboratory Medicine, Pisa University Hospital, Division of Pathology, Department of Translational Research and New Technologies in Medicine and Surgery, University of Pisa, Pisa 56126, Italy. [4]Department of Medicine, Vagelos College of Physicians and Surgeons, Columbia University Irving Medical Center, New York, NY 10032, USA. [5]Department of Molecular Pharmacology and Therapeutics, Vagelos College of Physicians and Surgeons, Columbia University Irving Medical Center, New York, NY 10032, USA. [6]Department of Pathology and Cell Biology, Vagelos College of Physicians and Surgeons, Columbia University Irving Medical Center, New York, NY 10032, USA. [7]Department of Urology, Vagelos College of Physicians and Surgeons, Columbia University Irving Medical Center, New York, NY 10032, USA. [8]Department of Systems Biology, Vagelos College of Physicians and Surgeons, Columbia University Irving Medical Center, New York, NY 10032, USA. [9]Department of Urology, Weill Cornell Medicine, New York, NY 10021, USA. [10]Department of Oncologic Pathology, Dana-Farber Cancer Institute and Harvard Medical School, 450 Brookline Ave, Boston, MA 02215, USA. [11]University of Oxford, Nuffield Department of Surgical Sciences, Oxford, UK. [12]These authors contributed equally: Hubert Pakula, Mohamed Omar, Ryan Carelli. ✉e-mail: mloda@med.cornell.edu

tumor- vs. non-tumor-bearing specimens, while benign epithelium does not display significant variability[8]. Furthermore, a stromal gene signature enriched in bone remodeling and immune-related pathways, largely overlapping with one derived from human xenografts that eventually metastasized[9], predicts metastases[8,9]. Importantly, prior analyses show that the stroma is composed of heterogenous and diverse cell populations whose roles in mediating disease progression have yet to be dissected[10]. In addition, whether the stromal micro-environment differs in the presence of diverse epithelial molecular subtypes of PCa remains to be determined. Therefore, genetically engineered mouse models (GEMMs) driven by different mutations and representing the different stages of prostate carcinogenesis can disentangle the complex stromal remodeling in PCa, shedding a light on complex interactions amongst epithelial, stromal, and immune components.

The *Tmprss2-ERG* (*T-ERG*) knock-in murine model[11] displays a mild epithelial phenotype and serves as a model of PCa initiation. The *Nkx3.1creERT2;Pten^f/f* (*NP*) mice[12–15], and the *Tg(ARR2/Pbsn-MYC)7Key* (*Hi-MYC*) GEMMs[16] represent prostatic intraepithelial neoplasia (PIN) with subsequent invasion. Advanced, aggressive, invasive adenocarcinoma and neuroendocrine prostate cancer (NEPC) is represented by the *Pb-Cre4 ^+/-;Pten ^f/f; Rb1 ^f/f;LSL-MYCN ^+/+* (*PRN)* model[17,18].

To investigate the tumor microenvironment (TME) in detail, particularly the mesenchymal cells associated with different epithelial lesions in these GEMMs, we are generating a comprehensive single-cell transcriptomic (scRNA-seq) compendium of the mouse PCa mesenchyme. This analysis contributes to the characterization of stromal cell subtypes with distinct expression profiles, which are likely regulated by key transcription factors orchestrating specific signaling pathways. We observe mesenchymal cell populations that are common across all GEMMs and wild-type (WT) mice, while others exhibit unique phenotypes that are associated with specific PCa drivers. Our investigations also cover the interactions within mesenchymal cell populations and between mesenchymal and other cell types, such as epithelial or immune cells. The regulons and interaction networks we identify suggest additional roles of the PCa stroma in mediating interactions within the tumor microenvironment. Moreover, there appears to be a preservation of cluster identity and spatial tissue architecture from murine models to human prostate cancer. Our ongoing work is illuminating distinct mesenchymal cell populations that may play varied roles in influencing the progression of PCa.

## Results

### Distinct stromal populations associated with different stages of prostate cancer

The stroma of PCa GEMMs differed significantly from that of their WT counterparts. In particular, stromal remodeling, characterized by an increase in collagen-rich extracellular matrix (ECM) deposition begins early in PCa carcinogenesis. Indeed, a significant expansion of the stromal compartment, as measured by image analysis. This expansion increases from models displaying PIN/microinvasion to the PRN model, which displays the greatest ECM deposition (Supplementary Fig. 1a, b). This finding highlights active remodeling of the stroma during tumor progression, suggesting that mesenchymal cells may change in function and composition during tumorigenesis.

To gain further insights into the composition and the function of the mesenchymal populations responsible for this stromal reaction, scRNA-seq profiles of 43,582 genes from 101,853 cells in 38 mice were collected using pooled single-cell suspensions of all lobes of the mouse prostate without a priori marker selection (Supplementary Data file 1). Cells of epithelial, lymphoid, endothelial, and neural origin based on the expression of canonical marker gene sets (Supplementary Data file 2) were excluded, yielding a dataset of 8574 mesenchymal cells. The

number of cells and transcripts from all models are shown in Supplementary Fig. 2a and Supplementary Data file 1. After correcting for batch effects and reducing dimension using a conditional variational autoencoder (VAE) (see "Methods"), we determined the different stromal cell types across all mouse models. To this end, we constructed a k-nearest neighbor graph in the VAE latent space using Euclidean metric, and clustered with the Leiden algorithm. This analysis revealed 12 stromal cell populations. Based on an analysis of cluster-cluster covariance and overlapping marker genes, three of these clusters were merged, while an additional cluster was removed as it had <5% of cells in WT mice. This resulted in a final number of eight distinct clusters (referred to as c0-c7). The distribution of the 8 mesenchymal clusters among the various GEMMs is shown in Fig. 1a. Some clusters are shared by GEMMs and WTs (c0-c2), while others are strongly enriched in particular mutant models (c3-c7) (Fig. 1b and Supplementary Fig. 2b).

Using ligand-receptor (L-R) interaction analysis, we found that the number and strength of signaling interactions between the stroma, epithelium, and immune compartments are greater in GEMMs compared to WTs (Fig. 1c). Specifically, outgoing signaling from the stroma of GEMMs is mainly mediated through interactions between subunits of collagen types I, III and IV on the stroma and their corresponding receptor, the collagen-binding integrin *Itga2/Itgb1* on the epithelium (Supplementary Data file 3). In addition, epithelium signaling to the stroma is mediated through the *Wnt4-Fzd1/Fzd2* and *Areg-Egfr* interactions (see Fig. 1d and Supplementary Data file 3). On the other hand, the only significant epithelial-stromal interaction in WTs is between the ligand *Gas6* in the epithelium and its receptor *Axl* in the stroma (Fig. 1d). Statistically significant signaling networks from the stroma to the epithelium in the prostates of WTs were not observed. This suggests that, while stroma to epithelium interactions do exist in normal prostates, they occur at a lesser frequency and strength compared to those found in GEMMs. Similarly, signaling from immune to stromal cells is significantly increased in GEMMs compared to WTs, mainly through the *Hbegf-Egfr* and *Mif-Ackr3* interactions (see Fig. 1d and Supplementary Data file 3). Overall, these results show that specific epithelial mutations do not just alter the stromal composition, but also induce significant changes in the inter-cellular communication networks in the microenvironment.

Since transcription factors (TFs) can play a role in cell lineage determination, knowledge of driving Gene Regulatory Networks (GRN) would improve cluster designations[19,20]. To this end, cis-regulatory network inference was used to identify potential regulons, consisting of a TF and its putative targets, driving either genotypes or clusters[21]. First, modules of highly correlated genes were identified, then pruned to include only those for which a motif of a shared regulator could explain the correlations. Subsequently, the activity of each regulon was scored in each cell and a set of regulons with different activity were identified in the eight mesenchymal clusters (Supplementary Fig. 2c). Differentially expressed genes (DEGs) in each cluster versus the remaining clusters (Supplementary Data file 4) were determined using the MAST approach, which employs a hurdle model to identify DEGs between two cell groups[22].

### Common mesenchymal clusters across different mouse models

Contractile marker genes including *Acta2, Myl9, Myh11*, and *Tagln*, and "muscle and smooth muscle cells contraction" by Reactome, are enriched in the c0 cluster. These cells also highly express *Mustn1, Angpt2*, and *Notch3*, suggesting a level of transcriptional complexity greater than previously suggested[23–25].

Interestingly, two distinct subpopulations of c0, named subclusters c0.1 and c0.2, are identified (Fig. 2a). Mesenchymal cells from c0.1 express myofibroblast marker genes such as *Rspo3, Hopx*, and *Actg2* (Fig. 2b, c)[24–26] while c0.2 overexpress pericyte markers (*Rgs5, Mef2c, Vtn, Cygb*, and *Pdgfrb*). Thus, the c0 cluster is composed of both bona fide myofibroblasts and pericytes. Although both sub-clusters are

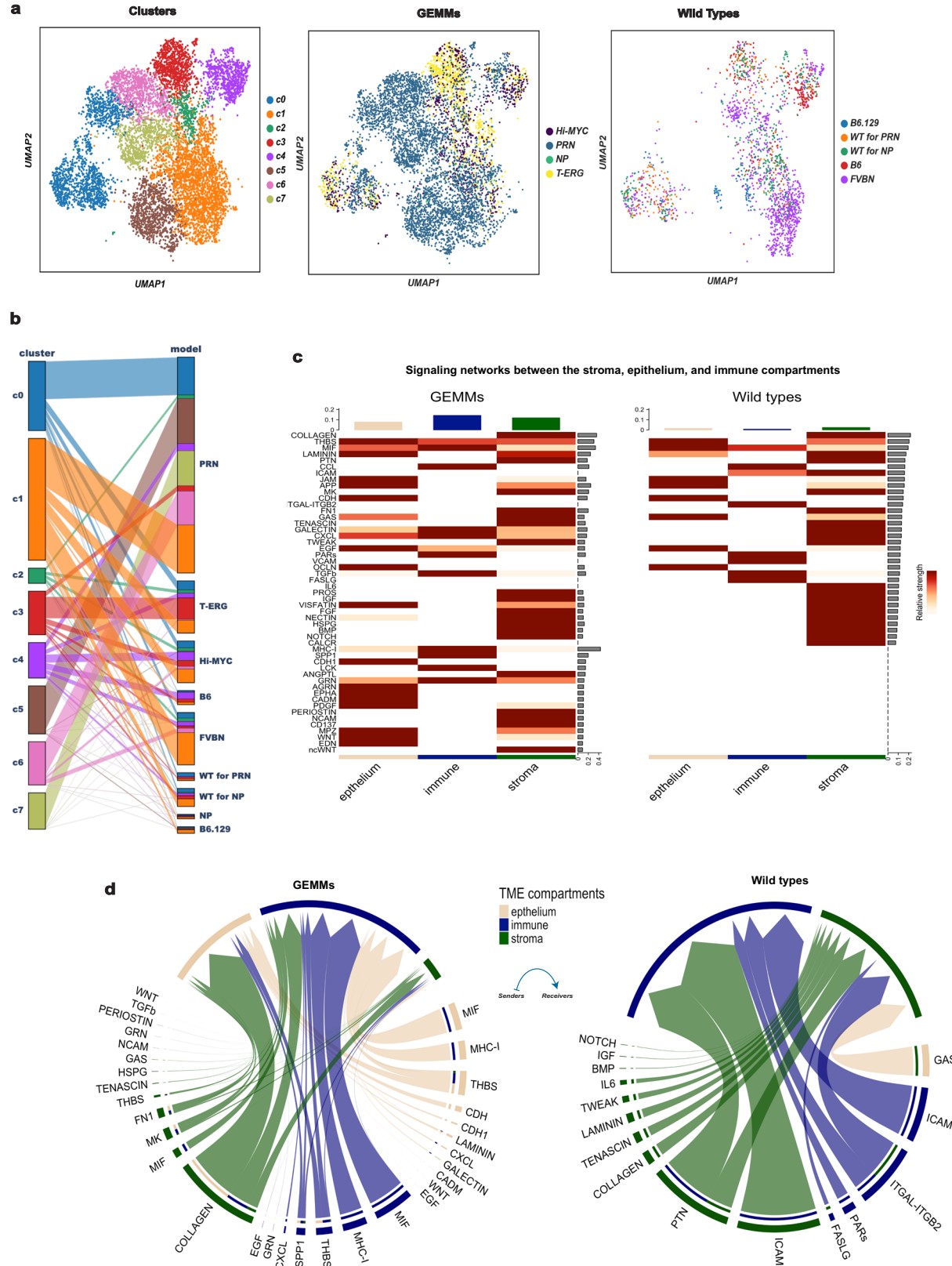

represented in all genotypes, sub-cluster 0.1 (c0.1) is predominantly found in *PRN* and sub-cluster 0.2 (c0.2) is enriched in *NP* (Fig. 2a). Regulon analysis confirmed the separation of these two sub-clusters, with c0.1 having a high expression of *Egr4, Crem*, and *Sox4* regulons, in contrast to c0.2 which has a high expression of *Egr2, Cebpa, Pparg, Klf2*, and *Klf4* (Fig. 2d).

Further investigation reveals the conservation of mesenchymal cells expressing innate immune response genes across different genotypes. Specifically, the common cluster c1 shows increased expression of *Sfrp1* and *Gpx3*, as well as major complement system components such as *C3, C7*, and *Cfh* compared to wild type. In addition, c1 shows a unique set of immunoregulatory and inflammatory

**Fig. 1 | Differential enrichment of stromal cell clusters in wild type versus genetically-engineered mouse models (GEMMs). a** Visualization of 8574 mesenchymal cells using Uniform Manifold Approximation and Projection (UMAP), color-coded based on their assignment to different clusters by graph-based clustering (left panel). The middle and right panels color-code these cells based on their model of origin (mutant vs. wild type). **b** Parallel categories plot showing the distribution of mesenchymal cell clusters (left) across the different mouse models (right). **c** Heatmap of the significant outgoing ligand-receptor (L-R) interaction patterns in the GEMMs (left) and wild-type (right) mice. The color bar represents the relative strength of a signaling pathway across cells. The top-colored bar plot represents the total signaling strength of each compartment by summing all the signaling pathways shown in the heatmap. The right gray bar plot indicates the total signaling strength of a particular pathway by summing all compartments presented in the heatmap. **d** Chord diagrams displaying the significant signaling networks between the stroma, epithelium, and immune compartments in mutants (left) and wild types (right). Each sector represents a distinct compartment, and the size of the inner bars represents the signal strength received by their targets. Up- and down-regulated signaling L-R pairs were identified based on differential gene expression analysis between mutants and wild types, with a log-fold change (logFC) of 0.2 set as a threshold. Communication probabilities for the L-R interactions were calculated after adjusting for the size of cell populations, and then aggregated on the signaling pathway-level. UMAP: Uniform Manifold Approximation and Projection.

genes (*Ccl11, Cd55, Ptx3,* and *Thbd*) as well as members of the interferon-inducible p200 family (*Ifi204, Ifi205,* and *Ifi207*) (Fig. 3a). Enrichment of GPX3 and C3 proteins in the stroma surrounding PIN and invasive tumor can be seen in Fig. 3b.

L-R interaction analysis revealed several communication networks from the epithelium to c0 and c1 stromal cells (Fig. 3c). These communications are mostly mediated through interactions between *Thbs1* found on epithelial cells exhibiting high expression of luminal marker genes (henceforth termed luminal-like cells) and *Sdc1, Sdc4,* and *Cd47* found on the stromal cells of c0 and c1, and through interactions between *Mif* expressed in luminal-like cells and *Ackr3* expressed mainly in c1 (see Fig. 3c and Supplementary Data file 3).

Notably, a distinct distribution of immune cell types is present in different mouse models (Fig. 3d). Immune signaling to c0 and c1 is dominated mainly by macrophages and to a lesser extent by dendritic cells (Fig. 3e). Macrophages signaling to c0 and c1 is mediated mainly through *Spp1* (expressed in macrophages) and integrins (expressed in c0 and c1). Specific Macrophages-c0 interactions are conducted through *Gzma* on macrophages and *Pard3* on c0, in contrast to specific macrophages-c1 interactions, which are mediated through the ligands *Mif* and *Hbegf on macrophages* and their corresponding receptors *Ackr3* and *Egfr* on c1 cells.

Through gene regulatory network analysis, several regulons were found to be specific to c0 and c1 stromal cells, including *Cebpalpha* and *Gabpb1*. In addition, other regulons govern inflammatory signaling systems such as *Nfkb1*, along with downstream genes involved in immune activation, which exhibit putative binding sites for these TFs (see Fig. 3a and Supplementary Fig. 2c).

Mesenchymal cells from c2 were found in all genotypes (Fig. 1b and Supplementary Fig. 2b). Components of the c-Jun N-terminal kinase (JNK) pathway were prominently expressed in this population. This is supported by high levels of *Ap-1* components including *Jun, JunB, JunD, Fos, FosD, FosB,* and *Fosl2*, activating factors (*Atf3*) (Fig. 3a). These were concomitant to increased expression of negative regulators of Erk1/2 such as *Dusp1, Dusp6,* and *Klf4* (Fig. 3a). GRN analysis revealed candidate TFs regulating MAPK superfamily such as *Atf3, Arid5a* and *Stat3* (Fig. 3a and Supplementary Fig. 2c). Mesenchymal cells in c2 also express both negative regulators of the Stat pathway and Stat-induced Stat inhibitors (SSI) (Fig. 3a). Interestingly, the expression of SSI family members is concomitant with strong expression of *Il6, Irf1*, which attenuate cytokine signaling. In addition, similar to c1, c2 interactions with immune cells in the TME are mediated mainly through the SPP1 and MIF signaling pathways (Fig. 3e).

## GEMM-specific mesenchymal clusters and communication amongst epithelial, stromal, and immune cells

Clusters c3 and c4 are predominantly enriched in *T-ERG, Hi-MYC,* and *NP* models. They express core components of the Wnt pathway including ligands, enhancers, negative regulators as well as master transcription factors (Fig. 4a). In situ validation of c3 and c4 markers by multiplex immunohistochemistry (mIHC) imaging confirms the

expression of WIF1 in *T-ERG* mesenchyme compared to the stroma of other mouse models, including *NP, Hi-MYC,* and *PRN* (Fig. 4b).

Signaling occurring between epithelial and immune cells and stroma in clusters c3 and c4 reveals GEMM-specific L-R interactions. For instance, in *T-ERG*, signaling from luminal-like cells to c3 and c4 stroma is mainly mediated through *Edn1-Ednrb, Thbs1-Sdc1/Sdc4,* and *Wnt4-Fzd1/Fzd2* interactions, while basal-like cells (epithelial cells with high expression of basal marker genes) and stromal signaling is mainly mediated through *Gas6-Axl, Col4a1-Sdc4,* and *Jag1-Notch2* interactions (see Fig. 4c and Supplementary Data file 3). On the other hand, luminal-stromal signaling in *Hi-MYC* is mediated solely by the *Mif-Ackr3* interaction, while basal-stromal signaling is conducted mainly by interactions between *Tgfb1* and its receptor *TGFbR1* (Supplementary Data file 3). Finally, in the *NP* mouse model, there is an increased activity of Wnt signaling from luminal-like and basal-like cells to c4 stroma, mainly through interactions between *Wnt4* and *Wnt7b* and their receptors on stromal cells including *Fzd2* and *Fzd5* (Supplementary Data file 3).

Immune-mediated signaling to c3 and c4 stroma also shows significant differences between the *T-ERG, Hi-MYC,* and *NP* models. For instance, in *T-ERG*, signaling from NK and cytotoxic T cells to the c3-c4 stroma are mediated mostly through *Fasl-Fas* and *Gzma-F2r* interaction (Fig. 4c) while in *Hi-MYC*, these networks involve mainly the *Lgals9-Cd44* interaction (Supplementary Data file 3). Similarly, signaling from dendritic cells to c3-c4 stroma is mediated through different L-R interactions across the three mouse models, with *T-ERG* characterized by increased activity of *Wnt11-Fzd1* and *Nectin1-Nectin3* interactions (Fig. 4c), while *Hi-MYC* and *NP* characterized by increased activity of the *Spp1-ltgav/ltga5* interaction (Fig. 4d).

Importantly, the immune tumor microenvironment of the *NP* model has a more prominent infiltration of monocytes/macrophages compared to the other models (Fig. 3d).

Several signaling networks between c3, c4 and other stromal cells in the TME especially the *PRN* clusters (c5-c7) were identified. The Wnt and non-canonical (nc) Wnt signaling pathways in particular are predominantly involved in mediating signaling from c3 and c4 (expressing several Wnt ligands like *Wnt5a, Wnt2,* and *Wnt4*) to the *PRN* clusters which express several Wnt receptors like *Fzd1* and *Fzd2* (Supplementary Fig. 3a).

Although both c3 and c4 have similar transcriptional and functional profiles, GRN analysis identified several candidate TFs underlying gene expression differences between the two clusters. For instance, while *Wnt*-stimulatory TFs, including *Sox9* and *Sox10*, drive c3, *Wnt*-repressive TFs such as *Foxo1* and *Peg3* are enriched in c4 (Fig. 4a and Supplementary Fig. 2c). Overall, these results suggest that the Wnt pathway plays an important, yet very complex role in these two clusters.

Cells belonging to clusters c5-c7 are associated with the NEPC mouse models, *PRN*[17,18]. Generally, cells in these clusters express cell cycle and DNA repair-related genes, neuronal markers, as well as a unique repertoire of collagen genes, Tgfβ activation, and again Wnt signaling. Specifically, these cells express high levels of the proliferative markers such as *Mki67* (Fig. 4a). They also show high

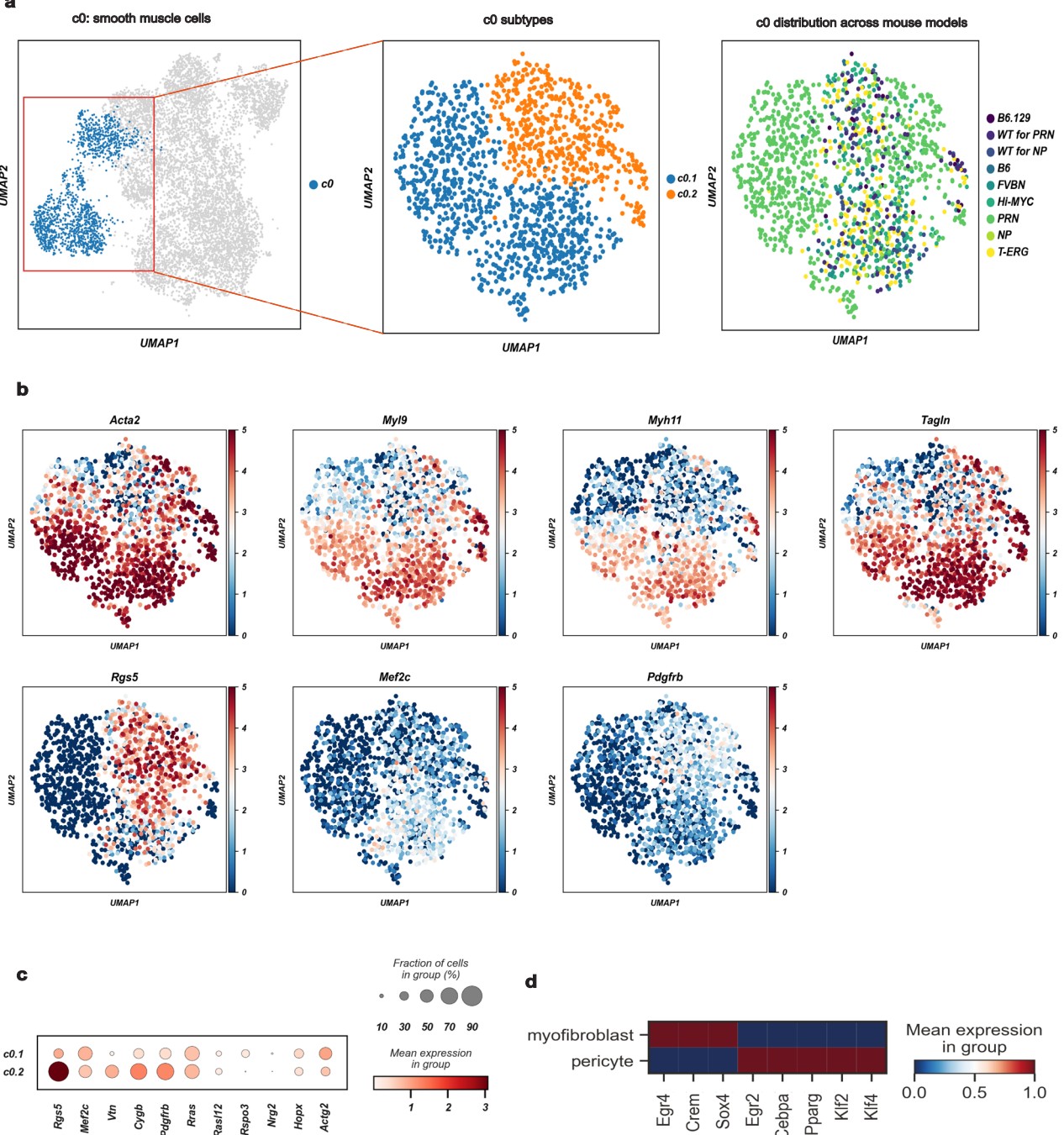

**Fig. 2 | A common cluster of contractile mesenchymal cells encompasses myofibroblasts and pericytes. a** Canonical myogenic and smooth muscle genes characterize c0 (*n* = 1401 cells) as contractile mesenchymal cells (left panel), but 2 subpopulations: c0.1 (*n* = 902 cells) and c0.2 (*n* = 499 cells) can be further subclassified (middle panel). Relative contribution of the different GEMMs and WTs to c0 is shown in the right panel. **b** UMAP projection of cells from c0 (*n* = 1401 cells), showing the expression of different myogenic and smooth muscle genes. *Acta2, Myl9, Myh11,* and *Tangl* mark myofibroblasts and pericytes, while *Rgs5, Mef2c,* and

*Pdgfrb* distinguish pericytes (c0.2). The color scale is proportional to the expression levels. **c** Dot plot showing the expression levels of genes that distinguish myofibroblasts (c0.1) from pericytes (c0.2). The color scale represents the mean gene expression in the cell groups. **d** The expression levels of regulons that distinguish myofibroblasts (c0.1) from pericytes (c0.2). The color scale represents the mean expression levels. GEMMs: genetically-engineered mouse models; UMAP: Uniform Manifold Approximation and Projection; WT: wild type.

expression of several downstream targets of the Wnt signaling pathway, Wnt receptors such as *Fzd1*, as well as Wnt-secreted decoy receptors *Sfrp*'s (Fig. 4a). Importantly, these clusters also highly express components of other signaling pathways such as *Tgfβ*-induced *Postn*, together with neuronal markers such as *Tubb3* (Fig. 4a). The complex stromal response in the *PRN* mouse model is also highlighted by a unique repertoire of upregulated collagen genes, such as *Col12a1,*

*Col14a1, Col16a1*, and metalloproteinase *Mmp19*, suggesting active remodeling in the TME (Fig. 4a). Several regulons driving these clusters involve transcription factors that generally define lineage in mesenchymal stem cells including *Gata6, Runx1*[27–29], *Gata2*[30], *Lhx6*, and *Snai3*[31,32] (Fig. 4a and Supplementary Fig. 2c).

The L-R interactions analysis revealed several signaling networks between the *PRN* stroma (c5-c7) and epithelial cells including luminal-

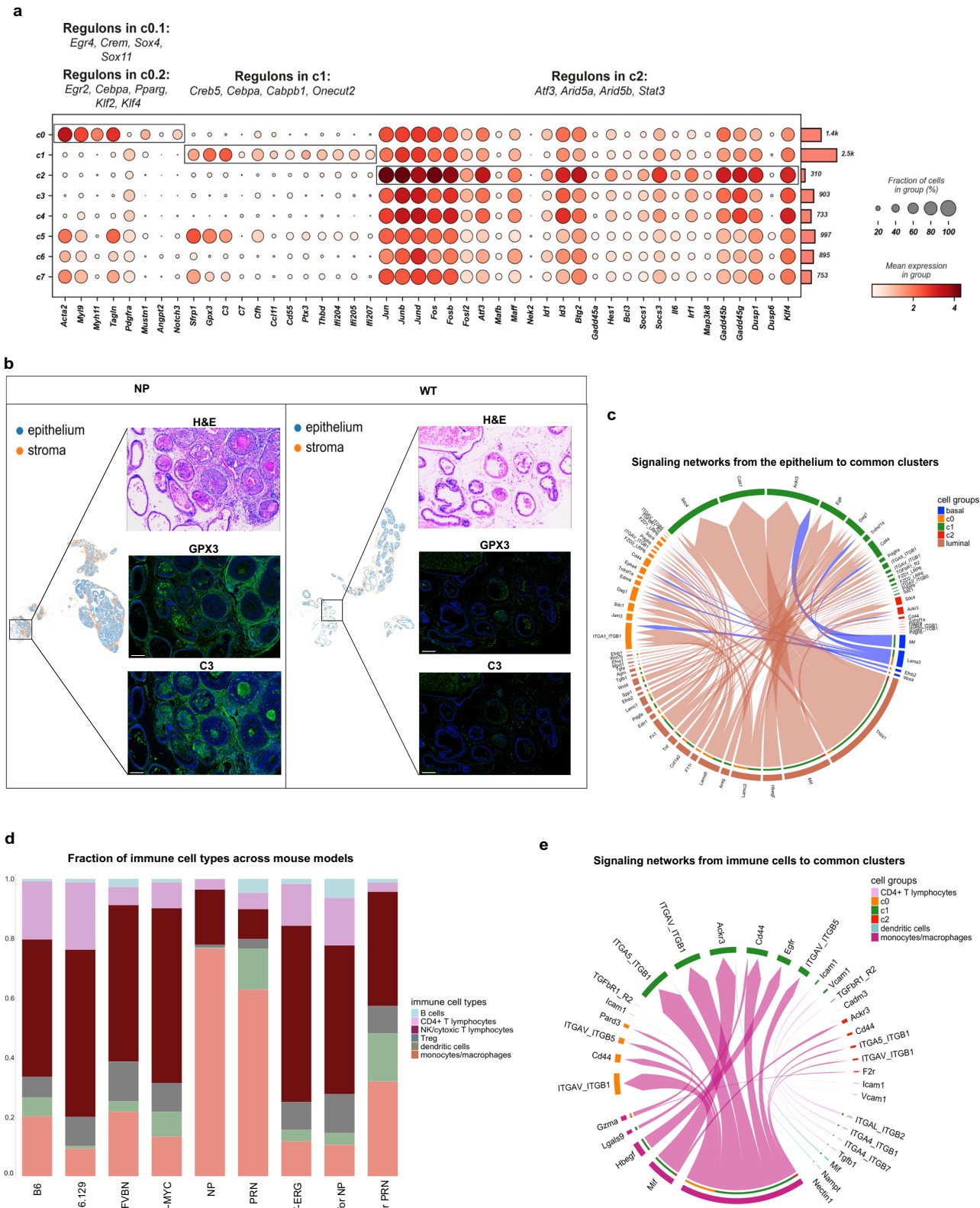

like, basal-like and NE-like (epithelial cells with high expression of NE marker genes) cells (Fig. 4d and Supplementary Data file 3). For instance, signaling from the luminal-like and basal-like epithelium to the stroma is mediated mainly through *Tgfβ1* and *Tnf* interactions with their respective receptors, as well as Wnt-mediated signaling.

In the immune TME, the inferred macrophage signaling to the *PRN* mesenchyme is driven mainly by interactions between *Spp1* and *Fn1*

(expressed in macrophages) and their receptors on c5-c7 including *Sdcs*, *Cd44* and integrins (Fig. 4d). Several L-R interactions are also inferred between the interleukins *1a/b* and *Tnf* with their respective receptors. Unlike other clusters, c5 appears to communicate via Il17a signaling with Tregs (Fig. 4d). In contrast, NK/cytotoxic T cells and the PRN mesenchyme communicate via the Interferon-gamma (IFNγ) signaling pathway (Fig. 4d). Stromal signaling through the Periostin pathway in particular is

**Fig. 3 | A functional atlas of the mouse prostate cancer mesenchyme. a** Dot plot showing the mean expression of marker genes for the common clusters (c0-c2). The boxes mark the clusters identified by each set of marker genes. The total number of cells in each cluster is indicated by the bar plot on the right. Significantly enriched regulons identified by gene regulatory network analysis are denoted on top of each boxed cluster. The color scale represents the mean gene expression in the cell groups. **b** Representative images of C3 and GPX3 overexpression in tumor desmoplastic stroma in the *NP* model (*n* = 3) (left panels) and matching WTs (*n* = 3) (one representative image for each model) (right panels). All images are at ×200 magnification with a scalebar of 300 μm. **c** Chord diagram of the significant ligand-receptor interactions from the epithelium to the common stromal clusters (c0-c2). Each sector represents a different cell population, and the size of the inner bars indicates the signal strength received by their targets. Communication probabilities were calculated after adjusting for the number of cells in each cluster. The displayed interactions are derived from all examined mouse models. Non-transformed basal cells and basal-like tumor epithelial cells are all grouped under the term basal. P-values are computed from a permutation test by randomly permuting the cell group labels (100 permutations), and then recalculating the communication probability. Only significant interactions (p-value < 0.05) are shown. **d** Bar plots showing the fraction of different immune cell types in the different mouse models. **e** Chord diagram showing the significant ligand-receptor interactions from different types of immune cells to the common stromal clusters c0-c2. Each sector represents a different cell population, and the size of the inner bars represents the signal strength received by their targets. Communication probabilities were calculated after adjusting for the number of cells in each cluster. The displayed interactions are derived from all examined mouse models. *P*-values are computed from a permutation test by randomly permuting the cell group labels (100 permutations), and then recalculating the communication probability. Only significant interactions (p-value < 0.05) are shown. H&E: hematoxylin & eosin staining; WT: wild type.

restricted to the PRN mesenchyme with few interactions involving c0 and c1 and no statistically significant interactions involving c3 and c4 (Supplementary Fig. 3b and Supplementary Data file 3).

*Postn* expression in c5-c7 is inversely correlated with *Ar* expression in the stroma. Interestingly *Ar* expression lowest in the PRN model compared to that in all other GEMMs (Fig. 5a, b). This reciprocal expression is found in all mouse models of advanced adenocarcinoma and NEPC including *PRN* (Fig. 5c), *PBCre4;Pten^{f/f};Rb1^{f/f} (DKO)* and *PBcre4;Pten^{f/f};Rb1^{f/f};Trp53^{f/f} (TKO)* models[33] (Supplementary Fig. 4), as well as in human samples (Fig. 5d). Generally, highest expression of POSTN/low AR is seen adjacent to the invasion front and surrounding foci of neuroendocrine differentiation (Fig. 5c, d). These findings are supported by Visium spatial transcriptomics (ST) profiling of prostate tissues performed in the PRN mouse model and respective WT (Fig. 5e).

These findings prompted us to assess whether *Postn*-positive stroma facilitates invasion, a characteristic of NEPC. Knockdown of Periostin in fibroblasts induces an over 2-fold decrease of mobility in a migration assay in 22rv1 cells overexpressing MYCN with additional *Rb1* knockdown to mimic the PRN model (Fig. 5f). In bulk RNA-seq data from a large cohort of well-characterized benign, locally advanced PCa, CRPC, and NEPC samples (https://shinyproxy.eipm-research.org/app/single-gene-expression), *POSTN* expression is significantly increased in a subset of CRPC and most NEPC patients compared to PCa and benign samples (Supplementary Fig. 5a).

### Functional validation of stromal cluster identity

These results suggest that specific epithelial mutations can shape the stromal microenvironment in PCa. To functionally validate these findings, we co-cultured normal fibroblasts from the *FVBN* mouse model with epithelial cells from the *T-ERG* and *PRN* models. We found that fibroblasts co-cultured with epithelial cells from *T-ERG* model tend to exhibit similar expression profiles to those found in c3 and c4 stromal cells, while those co-cultured with *PRN* epithelium exhibit expression profiles of the c5-c7 stromal clusters (Fig. 5g).

### Stromal transcriptional profiles are predictive of metastatic progression

The predictive and prognostic relevance of the *PRN*-derived clusters (c5-c7) was examined using gene expression profiles of primary tumor samples from a large cohort of PCa patients (*n* = 1239). The expression levels of the top positive and negative markers of the *PRN*-derived clusters were used as a biological constraint to train a rank-based classifier of PCa metastasis (see "Methods"). The resulting PRN gene signature consists of 13 up- and down-regulated gene pairs from the *PRN* mesenchyme (Supplementary Data file 5). In addition to its interpretable decision rules, this signature has a robust and stable performance in both the training (930 samples) and testing (309 samples) sets with an Area Under the Receiver Operating

Characteristic Curve (AUROC) of 0.69 and 0.70, respectively (Fig. 6a). Finally, the prognostic value of the signature was tested in the TCGA cohort which includes 439 primary tumor samples from PCa patients[34,35]. In this independent cohort, the PRN signature is significantly associated with progression-free survival (PFS)[34] using Kaplan–Meier survival analysis (log-rank *p*-value < 0.0001) (Fig. 6b), even after adjusting for Gleason grade in a multivariate Cox proportional hazards model (HR = 3.6, 95% CI = 1.2–11, *p*-value = 0.022) (Fig. 6c). Moreover, the *PRN* signature outperforms a cell cycle progression (CCP) signature when evaluated on the same testing set (Supplementary Fig. 6a). Unlike the *PRN* signature, the CCP signature is only significantly associated with PFS in univariate survival analysis (log-rank *p*-value = 0.001) (Supplementary Fig. 6b), and is not significant after adjusting for Gleason grade (HR = 4.5, 95% CI = 0.39–53) (Supplementary Fig. 6c). Overall, these results show that the *PRN*-derived mesenchymal cell clusters are associated with invasiveness, metastatic progression, and survival in PCa patients independent of Gleason grade.

### Human mesenchymal clusters in primary and metastatic tumors

Using the mouse scRNA-seq data as reference, the eight stromal clusters were mapped to the human scRNA-seq data[36]. These includes six ERG-positive (6990 mesenchymal cells) and three ERG-negative (1638 mesenchymal cells) patients. Notably, c3 is the predominant cluster in human stromal data (79% of total mesenchymal cells) (Fig. 7a), a finding attributed to the selection of ERG-positive cases. Notably, both c0 and c1 have transcriptional profiles similar to their murine counterparts, with c0 characterized by a high expression of myofibroblast marker genes (*ACTA2* and *MYL9*), and c2 cells having a high expression of *FOS* and *JUN* (Fig. 7b and Supplementary Fig. 5b). Notably, the transcriptional profiles of the GEMMs-specific stromal clusters are also preserved in the human data. For instance, stromal cells in c3-c4 have a high expression of genes involved in WNT signaling pathway including *WNT4* and *RORB* (Fig. 7b and Supplementary Fig. 5c). In contrast, the three *PRN*-associated stromal clusters (c5-c7) were found to be less abundant in human tumors (13% of total mesenchymal cells) compared to mouse specimens (31% of total mesenchymal cells), a finding explained by the absence of NEPC cases in the human primary tumor cohort. Nonetheless, these clusters still show transcriptional profiles similar to their mouse counterparts, with a high expression of *POSTN* and *SFRP4* (Fig. 7b and Supplementary Fig. 5d).

Analyses using scRNA-seq profiles of human PCa bone metastasis revealed transcriptional patterns similar to those present in the mesenchymal clusters from the *PRN* model (c5-c7), comprising more than 60% of bone stromal cells (Fig. 7c) and exhibiting a high expression of *POSTN* (Fig. 7d), together with osteoblasts marker genes, such as *BGN*, *RUNX2*, and *SPP1* (Fig. 6c, d). While the NEPC-related clusters are the predominant in the metastatic bone microenvironment, cells

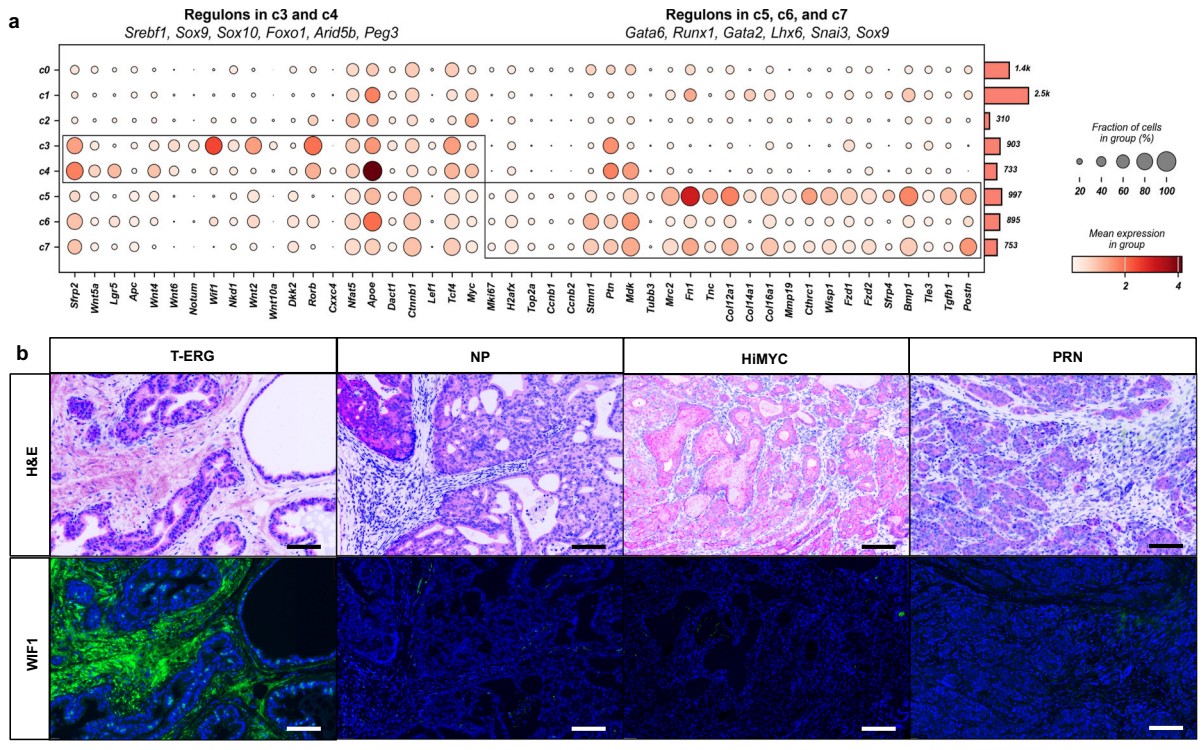

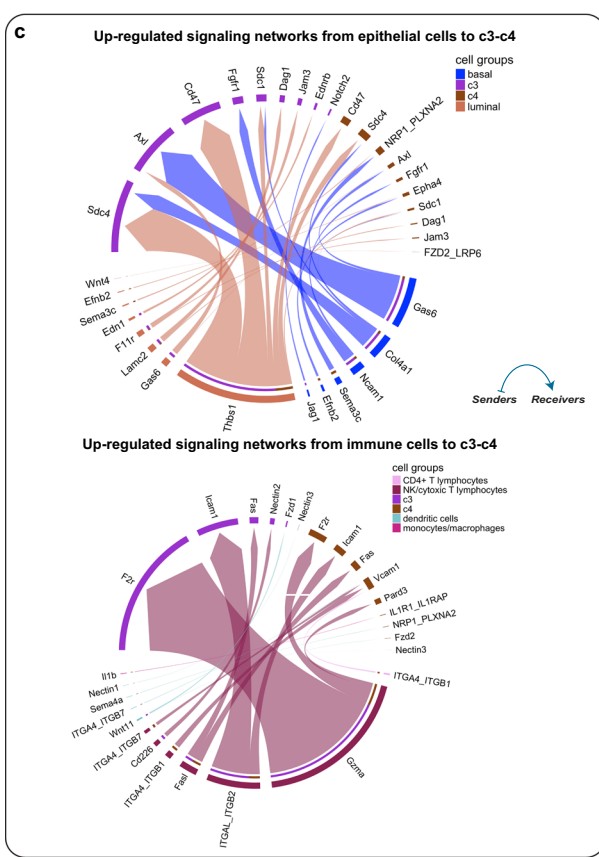

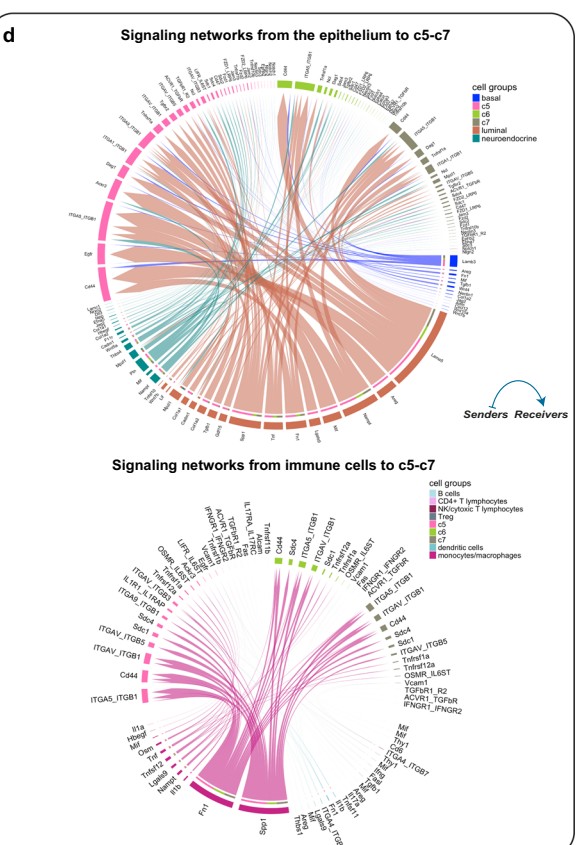

from c0 and c1 are also common, representing 9% and 27% of the total cells, respectively.

## Discussion

While different mutations in epithelial tumor cells partially explain the phenotypic and clinical heterogeneity of PCa, roughly one fourth of prostate tumors are genomically "quiet"[35], indicating that additional undiscovered determinants play a significant role in the biological behavior of PCa. Mesenchymal cells, which represent the predominant component of the microenvironment, have been suggested for decades to play a major role in this regard[37–39]. Recently, studies by Karthaus et al. and Crowley et al. described a detailed cluster analysis of

**Fig. 4 | GEMM-specific mesenchymal clusters define complex signaling pathways in the reactive stroma. a** Dot plot showing the mean expression of marker genes for model-specific clusters c3-c7. Boxes indicate the clusters marked by each marker gene set. The total number of cells in each cluster is indicated by the bar plot on the right. Significantly enriched regulons identified by gene regulatory networks are denoted on top of each boxed cluster. The color scale represents the mean gene expression in the cell groups. **b** Representative images of WIF1 expression in tumor desmoplastic stroma in the *T-ERG* (n = 4), *NP* (n = 3), *Hi-MYC* (n = 3), and *PRN* (n = 3) mouse models (one representative image for each model).

Magnification for all images ×200. Scalebar: 300 μm. **c** Chord diagrams showing the significantly upregulated ligand-receptor interactions from the luminal-like and basal-like epithelium (upper panel), and immune cells (lower panel) to c3 and c4 in *T-ERG* compared to *Hi-MYC* mouse models. **d** Chord diagrams showing the significant ligand-receptor interactions from the luminal, and neuroendocrine-like epithelium (upper panel), and also from immune cells (lower panel) to the *PRN*-associated clusters (c5-c7) in the *PRN* mouse model compared to its wild type. Communication probabilities were calculated after adjusting for the number of cells in each cluster. H&E: hematoxylin & eosin staining.

mesenchymal cells in the mouse prostate by scRNA-seq, revealing a level of complexity greater than that suggested previously[24,26]. Here, we analyzed in detail by scRNA-seq all mesenchymal cells utilizing all prostate lobes in the mouse prostate from several established GEMMs and corresponding WT mice. The significant and progressive increase in the mesenchymal cell component in increasingly aggressive GEMM models suggests a pivotal role of the stroma in tumor progression. We identified eight distinct stromal cell states that were defined by different gene expression programs and by underlying regulatory transcription factors. Three clusters represent fibroblast states that are common to all genotypes, and they display conserved functional programs across all stages of tumor growth. Five stromal cell states on the other hand, are specifically linked to defined epithelial mutations and disease stages, pointing to mutation-specific epithelial to stromal signaling.

There is growing evidence that innate immunity and inflammation play a role in prostate and other cancers[39–41]. While the focus of this study was not on immune cells, we found a cluster of mesenchymal cells conserved across all genotypes in prostate mesenchyme expressing genes associated with immunoregulatory and inflammatory pathways and driven by transcription factors such as Nfkb. Immune cells including tissue-resident macrophages are recruited and subsequently activated to modulate prostate tumorigenesis. In addition, stromal cells produce cytokines, chemokines, and components of complement protein pathways[42–44]. The complement system is an established component of innate immunity. Components of complement activation via the C3 alternative pathway were previously found to be activated by *KLK3* (a.k.a. PSA), with a special affinity for iC3b that in turn stimulates inflammation[45]. In addition, a pronounced expression of *Cd55* in common clusters, inhibits complement *C3* lysis[46]. The role of the complement as mediator of the stromal-immune crosstalk in c1 was also confirmed by the ligand-receptor analysis which showed significant interactions between *C3* and both ITGAM_ITGB2 and ITGAX_ITGB2 receptors in dendritic cells. This suggests that the expansion of cells expressing *C3* can stimulate innate immune response in the TME. Complex and bidirectional interactions between stroma and immune cells, mostly involve dendritic cells, monocyte/macrophages, and Tregs. Model-specific variations in the composition of the tumor immune microenvironment were seen, e.g., a prominent infiltration of monocytes/macrophages in the Pten model. Further functional analyses of those interactions will reveal how the stroma influences the response to immunotherapy in PCa[47–49].

Roughly half of prostate tumors have ETS translocations with *TMPRSS2* as the most frequent fusion partner[35], one of the earliest alterations in prostatic carcinogenesis[50–52]. Yet, genetically engineered mouse models driven by the TMPRSS2-ERG fusion display a minimal epithelial phenotype. Here, we found that induction of mesenchymal cell expansion is a significant early event in this model. We harmonized the eight murine clusters with human PCa cases sequenced using the same scRNA-seq approach. Strikingly, the mesenchyme associated with the TMPRSS-ERG translocation was conserved between mouse and human. Thus, epithelial ERG fusion in the mouse triggers early changes in the adjacent stroma, creating a TME that supports ERG-positive epithelial cells. Given the conservation of these mesenchymal clusters in humans, these findings shed a light on the role of this

prevalent alteration in the pathogenesis of prostate cancer. It will be important to determine the prevalence of these stromal cluster associated with TMPRSS-ERG in patients of African descent, where the prevalence of this translocation is low[53].

Stromal populations contribute to the structural and functional TME ecosystem through different autocrine and paracrine mechanisms. Among them, the stromal *AR* signaling cascade is known to influence prostate epithelial cells' behavior at different stages of development and carcinogenesis[37,54]. Stromal *AR* signaling may prevent invasion by maintaining a non-permissive TME for cell migration[55]. Indeed, loss of stromal AR was associated with upregulation of ECM-remodeling metalloproteinases (e.g., *MMP1*) and of *CCL2* and *CXCL8* cytokines, factors that promote invasion[55,56]. In the transgenic *Hi-MYC* and the testosterone+estradiol hormonal carcinogenesis models, stromal AR deletion favors prostate carcinogenesis[57]. In line with these observations, we show decreased mesenchymal *Ar* expression in the *PRN* model, which recapitulates late-stage PCa and progression toward neuroendocrine differentiation. Stromal *AR* may play a master role in committing and maintaining epithelial prostate cell identity in at least two ways. During development, its expression induces epithelial cells to differentiate into prostate cells, and during prostate carcinogenesis it prevents progression toward undifferentiated/neuroendocrine status.

Low expression of *Ar* in the *PRN* model was inversely associated with an increased expression of periostin (Postn), and in situ analyses confirmed that Postn-positive cells were enriched in areas of neuroendocrine differentiation. Stromal expression of periostin in PCa has been associated with decreased overall survival[58,59] and higher Gleason score[60]. We show that stromal cells expressing *Postn* confer invasive ability to poorly differentiated/NE carcinoma. The increased expression of *Postn* and of genes typical for the bone microenvironment (e.g., *Bgn, Runx2,* and *Spp1*) suggest that invasive PCa cells and the associated, invasion-primed mesenchyme modify the prostate TME to resemble that of bone, a common site of metastases in this malignancy. In this fashion, the primary site TME may pre-condition tumor cells for skeletal metastatic seeding. Importantly, we discovered shared characteristics between the stroma of the advanced/neuroendocrine GEMM and that of scRNA-seq stromal profiles from human bone metastases. Specifically, the bone stroma had a high frequency of *POSTN*+ cells. It is yet to be determined whether these cells were inherently present in the bone microenvironment or expanded as a result of the metastatic process. These cells were also characterized by the expression of genes involved in osteoblast differentiation and proliferation like *RUNX2, BMP2, IGF1,* and *IGFBP3*[61] together with Cadherin 11 (*CDH11*) previously found to induce PCa invasiveness and bone metastasis[62,63].

The role of complement is important not only in both modulating innate immunity but also invasion. A pronounced expression of *C1QA, B* and *C* was identified especially in models of advanced disease. *C1q* has been shown to promote trophoblast invasion[64] as well as angiogenesis in wound healing[65]. This was in line with our previously published stromal signature derived from laser capture-microdissected (LCM) mesenchyme adjacent to high-grade tumors that predicted lethality in an independent PCa cohort[8]. Three of the 24 signature genes were in fact *C1Q A, B,* and *C* suggesting that complement

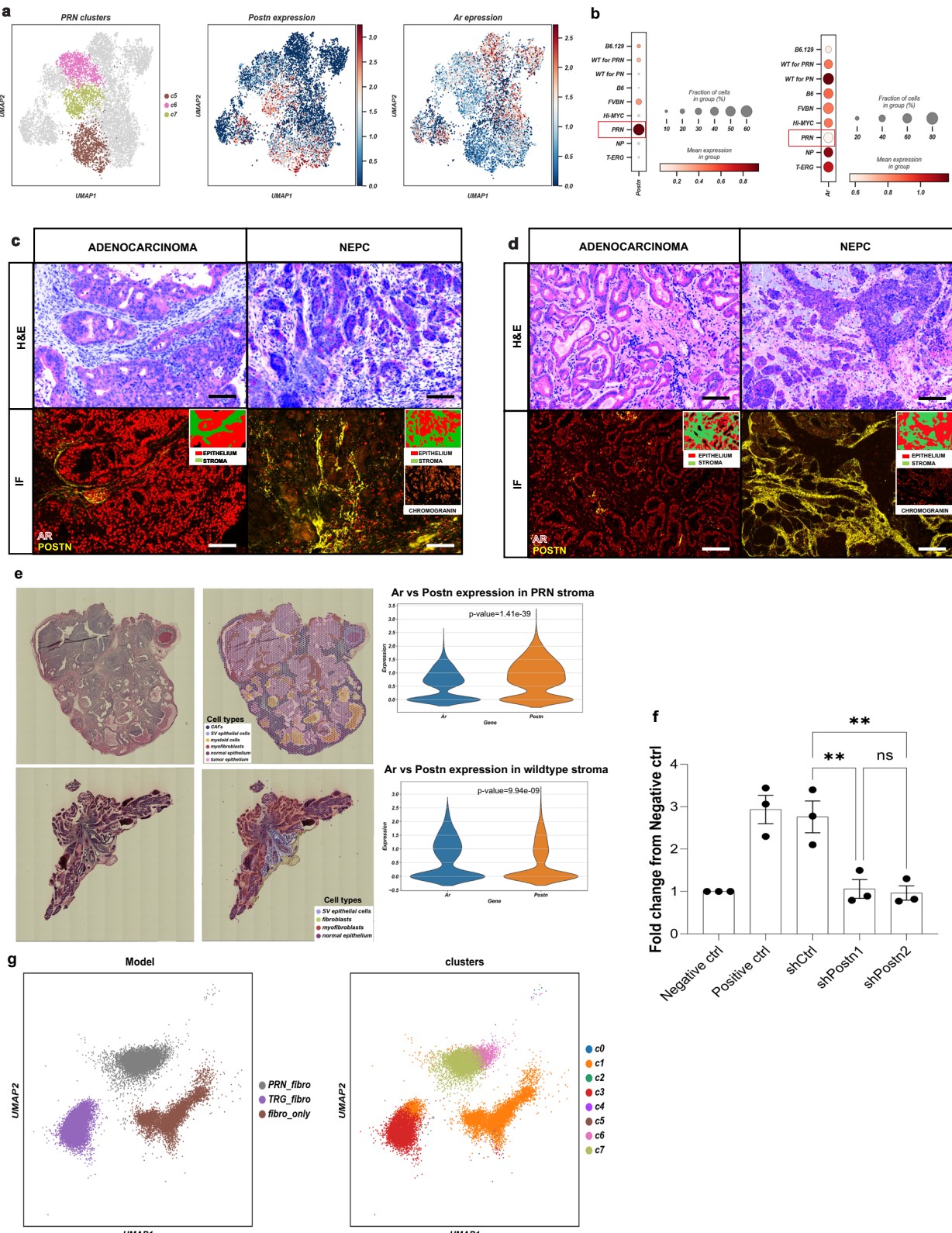

activation by the stroma plays a role in the invasive potential of aggressive prostate tumors (with diverse epithelial genetic alterations). The unexpected resemblance between PCa mesenchyme of locally aggressive tumors and that of bone metastases suggests that locally advanced PCa tumors prone to metastasize display a bone-like microenvironment. Since transcriptional profiles of stromal cells in aggressive models were conserved in the stroma of human localized

high-grade tumors[8], as well as in stroma of bone metastases in patients' biopsies[66], a broad set of cellular and molecular changes in the stromal cells may be either permissive or directly affect progression and metastatic disease.

Importantly, while our findings offer valuable insights on the role of the stroma in mediating PCa progression and invasiveness, they also show a strong translational relevance. For instance, we have used the

**Fig. 5 | Mesenchymal Periostin overexpression is associated with aggressive, neuroendocrine prostate cancer. a** UMAP projection of *PRN* clusters c5-c7 (*n* = 2645 cells) (left), along with the expression levels of *Postn* (middle) *and Ar* (right) in prostate mesenchyme (*n* = 8574 cells). **b** Dot plots of the mean expression of *Postn* and *Ar* in the different mouse models. The color scale represents the mean gene expression. **c, d** Multiplexed staining for a panel of proteins including Periostin, AR, and Chromogranin in *PRN* model (*n* = 3; one representative image of PRN is shown) (**c**) and human samples (*n* = 3; one representative image of human sample is shown) (**d**) showing high Periostin and low AR expression in stroma adjacent to neuroendocrine prostate cancer (NEPC) foci (right panel), and weak to moderate AR expression around in the stroma surrounding adenocarcinoma foci (left panel). Images are captured at ×200 and ×150 magnification for the PRN model and human cases, respectively with a scalebar of 300 μm. **e** Visium spatial transcriptomics of prostate tissue from the PRN mouse model and its wild type validates the expression of c5-c7 markers. Shown are the H&E-stained tissue sections (left) and overlay of the identified cell types based on gene expression (right). The violin plots compare the expression of *Ar* and *Postn* in the stroma. The *p*-values are derived

from a two-tailed t-test, and are as follows: Ar versus Postn expression in PRN stroma (*n* = 1638 cell spots): *p* = 1.41e−39, Ar versus Postn expression in WT stroma (*n* = 1166 cell spots): *p* = 9.94e−09. **f** Quantification of 22rv1 overexpressing MYCN and with Rb1 knockdown migration in Boyden chamber transwell assay. Data are expressed as mean ± SEM values of three independent experiments (*n* = 3, mean ± SEM). One-way ANOVA with Tukey's test, *$p < 0.05$, **$p < 0.01$, ***$p < 0.0003$. **g** Comprehensive analysis of scRNA-seq data obtained from primary prostate fibroblasts co-cultured with *T-ERG* and *PRN* epithelial cells. The UMAP plot on the left illustrates fibroblasts clustering, with each group annotated based on their genotypic identity indicating their co-culture conditions: *T-ERG* (*n* = 10529 cells), *PRN* (*n* = 7535 cells), or control group (*n* = 6921 cells). The UMAP plot on the right shows the fibroblast populations annotated according to their transcriptional similarities to the stromal subtypes identified in the scRNA-seq analysis of prostate tissues: c0 (*n* = 27), c1 (*n* = 8182), c2 (*n* = 19), c3 (*n* = 9327), c4 (*n* = 7), c5 (*n* = 8), c6 (*n* = 1066), and c7 (*n* = 6349). CAFs: cancer-associated fibroblasts; H&E: hematoxylin & eosin staining; IF: immunofluorescence; NEPC: neuroendocrine prostate cancer; UMAP: Uniform Manifold Approximation and Projection; WT: wild type.

scRNA-seq transcriptional profiles of the *PRN*-derived mesenchymal clusters (c5-c7) to develop a robust gene signature for predicting PCa metastases in a large cohort of patient samples with bulk transcriptomic profiles. This signature was also associated with worse progression-free survival in a separate cohort (TCGA) before and after adjusting for Gleason grade. In contrast, a cell-cycle progression (CCP) signature[67], did not have as robust a performance at predicting metastasis when tested on the same testing cohort. Furthermore, the CCP signature was not significantly associated with progression-free survival after adjusting for Gleason grade.

In summary, here we provide a molecular compendium of mesenchymal changes during PCa progression in genetically engineered mice that generalize to humans. Specifically, in the early phases of prostate carcinogenesis, we provide evidence that the TMPRSS-ERG translocation reprograms the mesenchyme which in turn may sustain progression. In advanced PCa models we found transcriptional mesenchymal programs linked to metastasis, some of them in common with the bone microenvironment to which PCa cells metastasize. The findings from those murine models have been validated and confirmed using publicly available and internally generated scRNA-seq data from ERG+ human tumors and PCa bone metastases. Collectively, these data from both mice and humans present clear evidence of significant shifts in stromal composition that accompany PCa progression, which are influenced by genotype-specific factors. These findings highlight the substantial role of mesenchymal changes as contributors to PCa progression and phenotypic diversity, emphasizing an impact that is more substantial than what has been detailed in existing literature.

## Methods
### Genetically engineered mouse models of prostate cancer
In this study, only males were utilized. All animals used in this study received humane care in compliance with the principles stated in the Guide for the Care and Use of Laboratory Animals (National Research Council, 2011 edition), and the protocol was approved by the Institutional Animal Care and Use Committee of Weill Cornell Medicine, Dana-Farber Cancer Institute and Columbia University Irving Medical Center. We focused on three models of prostate cancer that reflect the most common mutations in human localized disease, plus a fourth model that recapitulates the transition to NEPC. The choice of these models was also taken to reflect different stages of the disease.

Specifically, the TMPRSS2-ERG (*T-ERG*) (#RRID:MGI:5578645) fusion model has an N terminus-truncated human ERG together with an ires-GFP cassette into exon 2 of the mouse Tmprss2 locus[11,68], displays a minimal epithelial phenotype in the mouse, and was chosen since it represents the most frequent mutation in human prostate cancers[35]. These mice were bred on the same mixed genetic

background (FVB/N; Charles River Laboratories Stock CRL#: 207). *T-ERG* mice together with their WT counterparts were euthanized and analyzed at the age of 6 months. *Pten knock-out* (*NP*) (Jackson Laboratory, #RRID:IMSR_JAX:033751)[14,15] mice develop high grade PIN with areas of invasion. To obtain *Nkx3.1creERT2;Pten^{f/f}; EYFP^{f/f}* (*NP*) *Nkx3.1^{creERT2}* driver (Jackson Laboratory #RRID:IMSR_JAX:032753) was crossed to the conditional allele for *Pten* (Pten^{flox/flox})[12] Jackson Laboratory #RRID:IMSR_JAX:006440) with loxP. For induction of Cre activity in *NP* mice, tamoxifen (Sigma Cat #T5648) (or corn oil alone) was delivered by IP injection (225 mg/kg) for 4 consecutive days, to mice at 2 months of age. Six months later *NP* mice were sacrificed and analyzed. *Hi-MYC*[16] shows both PIN and microinvasion. FVB *Hi-MYC* mice (strain number 01XK8, #RRID:MGI:5486199), expressing the human c-MYC transgene in prostatic epithelium, were obtained from the National Cancer Institute Mouse Repository at Frederick National Laboratory for Cancer Research. These mice were bred on the same mixed genetic background (Charles River Laboratories Stock CRL#: 207). *Hi-MYC* mice together with their WT counterparts were euthanized and analyzed at the age of 6 months. *PRN* mice recapitulates the transition to NEPC[18]. In brief, mice carrying the CAG-LSL-MYCN human transgene at the *Rosa26* locus (LSL-MYCN+/+)[69] were crossed with mice expressing Cre recombinase under the control of rat *Probasin*, a prostate-specific promoter (Jackson Laboratory, #026662, RRID:IMSR_JAX:026662), along with *Pten* homozygous floxed alleles (*PbCre^{+/−}; Pten^{f/f}*). Resulting males that carried the *MYCN* transgene, Cre recombinase, and *Pten* floxed alleles were crossed with females carrying *Rb1* homozygous floxed alleles (Jackson Laboratory, #026563, RRID:IMSR_JAX:026563). Prostate-specific Cre expression results in removal of LSL cassette by Cre and human N-Myc expression driven by the chicken actin promoter. Simultaneously, Cre recombinase converts the *Pten* and *Rb1* floxed alleles to knockout alleles in the mouse prostate. All lines of mice were bred on the same mixed genetic background (C57BL6/129 × 1/SvJ) and have been previously described[68]. *PRN* mice together with their *WT* counterparts were euthanized and analyzed at the age of 8 weeks. In addition, non-littermate WT mouse strains, *FVB/N* Charles River Laboratories, CRL#RRID:IMSR_CRL:207), *C57BL/6* (Jackson Laboratories RRID:IMSR_JAX:000664) and *B6129SF2/J* (Jackson Laboratory RRID:IMSR_JAX:101045) were used. All mice were housed in a specific pathogen-free (SPF) environment maintained at 72 ± 2 °F (21.5 ± 1 °C), relative humidity between 30% and 70%, with a 12/12 h dark-light cycle (lights on at 7:00 a.m.), with free access to food and water. For all in vivo studies, limits to tumor size of 10% of the average mouse body weight or 2 cm in any one dimension were applied as approved by the Weill Cornell Medical College IACUC. These limits were not exceeded in any studies.

The number of GEMMs and their WTs and/or littermates are provided in Supplementary Data file 1.

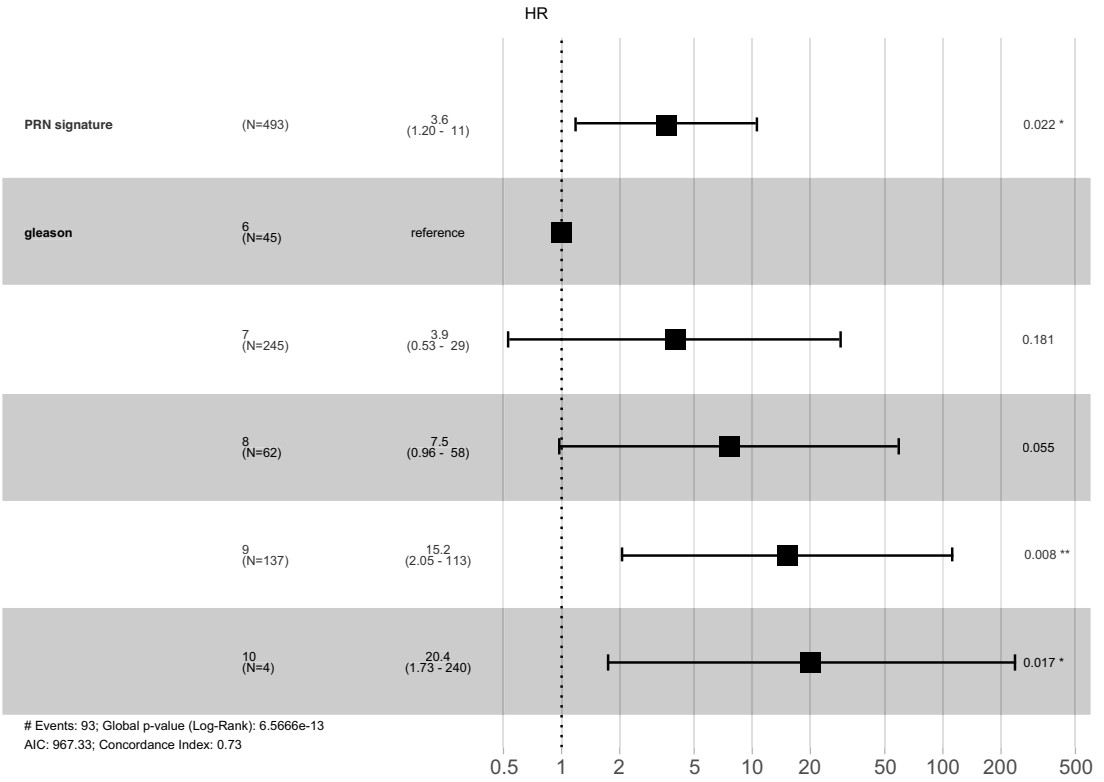

**Fig. 6 | The transcriptional profiles of the PRN-derived clusters are predictive of metastatic progression in prostate cancer. a** Receiver Operating Characteristics (ROC) curve displaying the predictive performance of the PRN signature for metastasis in both the training ($n = 930$) and testing ($n = 309$) data. The signature was trained and validated on bulk expression profiles derived from primary tumor samples of PCa patients. AUC: area under the ROC curve. **b** Kaplan–Meier survival plot illustrating the differences in progression-free survival (PFS) between patients predicted to have metastasis (predicted PFS:1) and those predicted to be metastasis-free (predicted PFS:0) in the TCGA prostate adenocarcinoma cohort

($n = 439$). The x-axis represents survival time in months. The observed difference in survival is statistically significant with a *p*-value of <0.0001, assessed using the log-rank test. **c** Forest plot for multivariate Cox proportional hazards model depicting the hazard ratio (HR) (central black square) and 95% confidence interval (CI) (horizontal lines) for both the *PRN* signature ($p = 0.02$) and different Gleason grades (Gleaon 7 $p = 0.18$, Gleason 8 $p = 0.06$, Gleason 9 $p = 0.01$, Gleason 10 $p = 0.02$). Significance, indicated by an asterisk, is based on the *p*-value from the Wald test in the Cox model (**p*-value < 0.05).

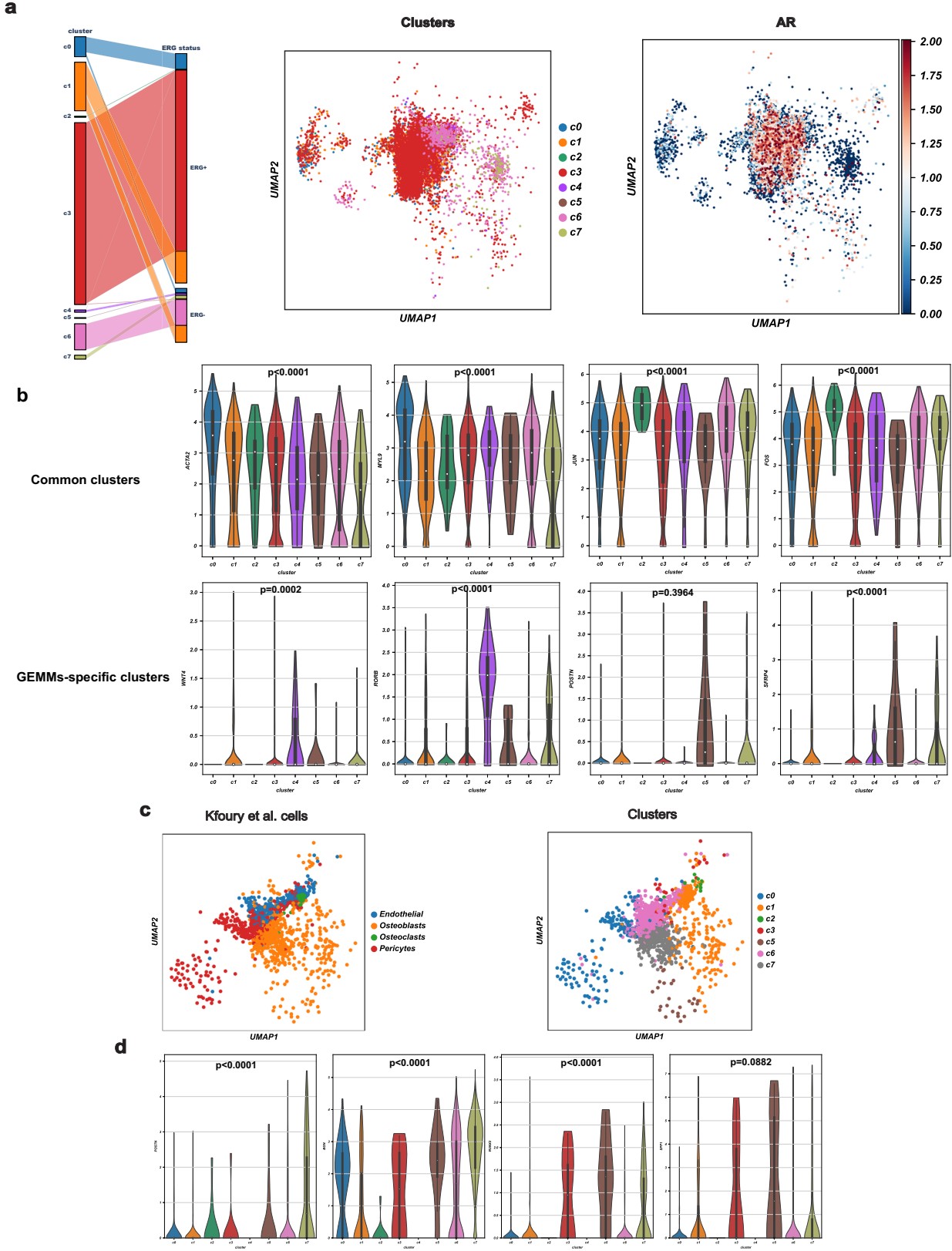

## Description of human prostate cancer specimens

Human prostate tissue specimens were obtained from patients undergoing radical prostatectomy at Weill Cornell Medicine under Institutional Review Board approval with informed consent (WCM IRB #1008011210, #1302013582). The study included a total of 13 subjects, all of whom were male. No blinding, randomization, or exclusion criteria were applied. Of these, 9 samples (comprising 3 ERG-negative and 6 ERG-positive cases) were utilized for single-cell RNA sequencing studies, while 4 samples were employed for the mIHC Vectra Polaris staining. The clinical and molecular characteristics of these patients are provided in Supplementary Data file 6. Immediately after surgical removal, the prostate was sectioned transversely through the apex,

**Fig. 7 | Analysis of human scRNA-seq data suggests the relevance of prostate mesenchyme in human PCa pathobiology. a** Parallel categories plot showing the relationship between the mesenchymal clusters and ERG status (left). UMAP projection of the eight mesenchymal clusters (n = 8628 cells) in the human scRNA-seq data (center) and *AR* expression across the human mesenchymal clusters (right). **b** Violin plots depicting the expression of marker genes for stromal clusters in the human scRNA-seq data, derived from n = 9 patients, encompassing a total of 8628 individual cells. The width of the violins at different values represents the density of the data. The embedded box plots display the median of the data (white dot), the bounds of the box represent the 25th and 75th percentiles (interquartile range), and the data within these bounds represent the minima and maxima of the non-outlying data. P: *p*-value derived from Welch's t-test comparing the expression of each marker gene in each corresponding cluster to its expression in the remaining clusters and are as follows: ACTA2: $p < 0.0001$, MYL9: $p < 0.0001$, JUN: $p < 0.0001$,

FOS: $p < 0.0001$, WNT4: $p = 0.0004$, RORB: $p < 0.0001$, POSTN: $p = 0.39$, SFRP4: $p < 0.0001$. **c** UMAP of the selected cell types from the bone metastasis scRNA-seq data derived from Kfoury[66] (left) and their corresponding annotation using the eight mesenchymal clusters definition (middle). **d** Violin plots showing the mean expression of *POSTN, RUNX2, SPP1,* and *BGN* across the mesenchymal clusters in the scRNA-seq bone metastasis cohort from Kfoury[66], derived from n = 9 bone samples, encompassing a total of 1872 individual cells. The embedded box plots display the median of the data (white dot), the bounds of the box represent the 25th and 75th percentiles (interquartile range), and the data within these bounds represent the minima and maxima of the non-outlying data. P: *p*-values resulting from Welch's t-test comparing the expression of each marker gene in c5-c7 versus the remaining clusters and are as follows: POSTN: $p < 0.001$, BGN: $p < 0.0001$, RUNX2: $p < 0.0001$, SPP1: $p = 0.08$. UMAP: Uniform Manifold Approximation and Projection.

mid, and base[70]. Tissue for scRNA-seq was placed in RPMI medium with 5% fetal bovine serum (FBS) on ice, and quickly transported for single-cell RNA sequencing. A small portion of the regions of interest, including the areas selected for single-cell RNA sequencing, index lesion, and contralateral benign peripheral zone, was concomitantly frozen in optimal cutting temperature (OCT) compound, cryosectioned, and a rapid review was performed by a board-certified surgical pathologist (BR) to provide a preliminary assessment on the presence of tumor, normal epithelium, stroma near and away from the tumor. Adjacent tissue was processed by formalin fixation and paraffin embedding, followed by sectioning, histological review, histochemistry (trichrome stain), and immunostaining[71].

### Isolation of single cells for RNA-Seq

Dissociated murine prostate cells were prepared as described previously[72]. Briefly, mouse prostate tissues were digested in Advanced DMEM/F12/Collagenase II (1.5 mg/ml)/Hyaluronidase VIII (1000 u/ml) (Thermo Fisher Scientific) plus 10 μM Y-27632 (Tocris) for 1 h at 37 °C with 1500 rpm mixing, continuously agitated. Subsequently, after centrifuging at $150 \times g$ for 5 min at 4 °C, digested cells were suspended in 1 ml TrypLE with 10 μM Y-27632 and digested for 15 min at 37 °C and neutralized in aDMEM/F12/FBS (0.05%). Dissociated cells were subsequently passed through 70 μm and 40 μm cell strainers (BD Biosciences, San Jose, CA) to obtain a single cells suspension. Samples were resuspended in 1x PBS and sorted by Flow Cytometry (Becton-Dickinson Aria II and/or Becton-Dickinson Influx) for 4',6-diamidino-2-phenylindole (DAPI) to enrich for living cells.

Similarly, human prostate tissues were first digested in aDMEM/F12/Collagenase II (1.5 mg/ml)/Hyaluronidase VIII (1000 u/ml; Thermo Fisher Scientific) plus 10 μM Y-27632 (Tocris) for 1 h at 37 °C with 1500 rpm mixing, continuously agitated. Subsequently, after centrifuging at $150 \times g$ for 5 min at 4 °C, digested cells were suspended in 1 ml TrypLE with 10 μM Y-27632 and digested for 15 min at 37 °C and neutralized in aDMEM/F12/FBS (0.05%). Dissociated cells were subsequently passed through 70 μm and 40 μm cell strainers (BD Biosciences, San Jose, CA) to get single cells. Samples were resuspended in 1x PBS and sorted for DAPI to enrich living cells.

Barcoded cDNA libraries were created from single-cell suspensions using the Chromium Single Cell 3' Library and Gel Bead Kit, and Chip Kit from 10x Genomics[73], according to manufacturer recommendations. Briefly, depending on the GEMMs and human samples used in this study, 8000–16,000 cells were targeted for 3' RNA library preparation, multiplexed in an Illumina NovaSeq 6000, and sequenced at an average depth of 25,000 reads per cell.

### Quantification and preprocessing of single-cell RNA sequencing data

Expression matrices were generated from raw Illumina sequencing output using CellRanger. Bcl files were demultiplexed by bcl2fastq, then reads were aligned using the STAR aligner[74] with the default

parameters. All data collected from mouse models were aligned to GRCm38 reference transcriptome. To identify cells with trans-gene expression, we indexed and aligned to human ERG and GFP from the *T-ERG* model, human MYC from the *Hi-MYC* model, and human MYCN from the *PRN* model. Human data were aligned to GRCh38. Alignment quality control was performed using the default CellRanger settings. Expression matrices from the different mouse models were converted to AnnData objects and concatenated into a single count matrix using the Scanpy library (version 1.5) in Python (version 3.8)[36]. Similarly, the expression matrices from the nine human samples were concatenated into a single count matrix. The raw mouse and human scRNA-seq count matrices were preprocessed as follows: cells with low UMIs (unique molecular identifiers) count (<400) and low number of expressed genes (<300) were removed. Subsequently, genes that were expressed in three or fewer cells and cells containing more than 20% mitochondrial transcripts were removed after visualizing the distribution of fraction of counts from mitochondrial genes per barcode[75]. Contributions from total count, mitochondrial count, and cell cycle were corrected by linear regression. The resulting matrix was then log1p transformed[75]. Finally, the top 4000 genes were selected based on the coefficient of variation according to the method described in ref. 73, and genes were scaled to a mean of zero and unit variance[75].

### Embedding of scRNA-seq expression matrix by deep generative modeling

We computed batch-corrected embeddings as follows. We fit our data using a conditional variational autoencoder[76]. Specifically, we used the negative binomial counts model included in the single-cell variational inference (scVI) Python package[77]. We model (a) a nuisance variable that represents differences in capture efficiency and sequencing depth and serves as a cell-specific scaling factor, and (b) an intermediate value that provides batch-corrected normalized estimates of the percentage of transcripts in each cell that originate from each gene. Our model is implemented in Python using the PyTorch library (v1.7.0)[78] and was run on a NVIDIA RTX A4000 GPU.

### Clustering and data visualization

A nearest neighbor graph was constructed with Euclidean metric from the batch-corrected scVI embeddings, then cells were partitioned by the Leiden clustering algorithm[79,80]. Partition-Based Graph Abstraction (PAGA) was computed from the Leiden partition[81] and was used to initialize the Uniform Manifold Approximation and Projection (UMAP) algorithm which projected the data into 2D space[82].

### Identification of stromal cells

For both the mouse (101,853 cells) and human (83,080 cells) scRNA-seq datasets, we excluded cells of lymphoid, endothelial, and neural origin based on Leiden clustering at resolution 1.0 and the expression of associated lineage markers (Supplementary Data file 2)[24,26,83–85]. The resulting mesenchymal datasets for the mouse and human scRNA-seq

data included 8574 and 8628 cells, respectively. These mesenchymal cells were then clustered using the Leiden algorithm to identify different mesenchymal sub-clusters. Specifically, at resolution 0.05, the Leiden clustering reflected the separation of *Mesenchymal* and *Smooth Muscle Cells/Myofibroblasts* subtypes. We increased resolution in increments of 0.05, inspecting the biological plausibility of new clusters until resolution 0.6, after which higher resolution produced new clusters with differences dominated by noise[79].

## Identification and annotation of immune cell types
In the mouse scRNA-seq data, cells from the immune compartment (42,431 cells) were also clustered using the Leiden algorithm. The resulting clusters were then annotated to different immune cell types based on the expression of known markers genes. These included B cells (expressing *Cd79a, Cd79b, Cd74, Cd19*, and *Cd22*), CD4 + T lymphocytes (expressing *Cd4, Cd2, Cd28*, and *Trac*), NK or cytotoxic T cells (expressing *Xcl1, Nkg7, Gzmb, Klrc1*, and *Klrc2*), Tregs (expressing *Foxp3, Ctla4, Tnfrsf4*, and *Tnfrsf18*), dendritic cells (expressing *Ccl17, Ccr7, Xcr1*, and *Cd207*), and monocytes or macrophages (expressing *Cd68, Cd74, Cxcl2*, and *Lgals3*).

## Differential expression testing
For differential expression (DE) testing, we used a two-part generalized linear model (hurdle model), MAST, that parameterizes stochastic dropout and the characteristic bimodal distribution of single-cell transcriptomic data[22]. DE was performed by comparing the gene expression profiles of cells from each cluster to pooled cells from all other clusters (Supplementary Data file 4). Using the default parameters, the DE analysis was limited to genes which show on average at least 0.25-fold difference between the two cell groups, and only genes that are detected in at least 10% of cells in either groups.

## Gene regulatory network inference (GRN)
Gene regulatory network activity was inferred from the raw counts matrix by pySCENIC (v0.10.3)[21]. Specifically, coexpression modules between transcription factors (TFs) and their candidate targets (regulons) were inferred using the Arboreto package (GRNBoost2) and pruned for motif enrichment to separate indirect from direct targets[21,86]. The activity of each regulon in each cell was then scored using the Area Under the ROC curve (AUC) calculated by the *AUCell* module from pySCENIC package[21,86]. Cluster-specific regulons were identified as those with AUCell Z-score >1 for each mesenchymal cluster.

## Ligand-receptor analysis
We performed ligand-receptor (L-R) interaction analysis using CellChatDB and CellChat R tool (version 1.1.3) to predict cell–cell interactions within the tumor microenvironment[87]. Cell communication networks were inferred by identifying differentially expressed ligands and receptors between the different mesenchymal clusters, immune cell types, and the epithelium. The probabilities of these interactions on the ligand-receptor level were computed using the default 'trimean' method setting the average expression of a signaling gene to zero if it is expressed in less than 25% of the cells in one group. Notably, we corrected for the effect of population size (number of cells) when calculating the interaction probabilities. In addition, we summarized the L-R interaction probabilities within each signaling pathway to compute pathway-level communication probabilities as described in ref. 87. Cell–cell communication networks were then aggregated by summing the number of interactions or by averaging the previously calculated communication probabilities. To compare the signaling patterns between mutants and wild types, we performed differential expression analysis between all the mutants versus wild types in each of the three compartments (stroma, epithelium, and immune). Upregulated ligands and receptors were identified if each had a log-fold

change (logFC) above 0.1 in the senders and receivers, respectively. Finally, we extracted the mutant-specific L-R pairs as those with upregulated ligands and receptors in the mutants compared to wild types and vice versa. In this analysis, we used a *p*-value threshold of 0.05 to determine significant interactions.

## Label transfer from mouse to human scRNA-seq data
To transfer the stromal cluster labels from the mouse to human data, human gene symbols were converted to their mouse counterparts then both datasets were subset to overlapping genes. Label transfer was performed using '*ingest*'[36] which maps the labels and embeddings fitted on an annotated reference dataset to the target one. Specifically, we used the scRNA-seq data from the mouse *T-ERG* model as reference for the human ERG-positive cases and those from the remaining mouse models as reference for the human ERG-negative cases. Finally, we computed the ranking of differentially expressed genes in each cluster versus the remaining ones using *t*-test.

## Processing of human bone metastases scRNA-seq data
The raw count matrix of the scRNA-seq dataset previously reported by Kfoury et al.[66] was retrieved from the Gene Expression Omnibus (GEO). This dataset included 25 bone metastasis samples derived from PCa patients, of which 9 samples were derived from solid metastasis tissue. Further analysis was limited to these 9 samples (16,993 cells). The data was preprocessed by keeping cells with at least 200 expressed genes and less than 15% mitochondrial transcripts (16,536 cells). Subsequently, cells were normalized by the total counts over all genes followed by log scaling and regressing over the total counts per cell and percentage of mitochondrial genes to reduce unwanted variation. The top 4000 highly variable genes were selected and the resulting matrix was then scaled to unit variance and zero mean. Since this particular analysis was intended to explore the transcriptional and functional similarities between the primary tumor stroma and the stroma of bone metastasis, we further limited the analysis to the cells previously annotated by the authors as osteoblasts, osteoclasts, endothelial cells, and pericytes (1872 total cells). Finally, the embeddings and stromal cluster labels were projected onto this dataset using the mouse stroma scRNA-seq dataset as reference and following the same steps mentioned above.

## Development of the PRN signature to predict metastasis in prostate cancer patients
We collected and curated gene expression profiles from different datasets comprising 1239 primary tumor samples from PCa patients with information about metastatic events. These datasets included six publicly available datasets (GSE116918[88], GSE55935[89], GSE51066[90], GSE46691[91–93] GSE41408[94], and GSE70769[95]), together with a seventh dataset available from Johns Hopkins University, referred to as the natural history cohort[96] and https://zenodo.org/doi/10.5281/zenodo.7452769[97]. The expression profiles from each dataset were normalized, log2-scaled, then z-score transformed (by gene) separately. Subsequently, we mapped probe IDs to their corresponding gene symbols and kept only the genes in common between all datasets (12,761 genes).

The 1239 samples were joined together then split into 75% training (n = 930) and 25% testing (n = 309) using a stratified sampling approach to ensure an equal representation of important variables including the original datasets, Gleason grade, age, tumor stage, and prostate-specific antigen (PSA) levels. Quantile normalization was applied to both the training and testing sets separately. The training set was used for training a classifier that can predict metastasis using the k-top scoring pairs (k-TSPs) algorithm, which is a rank-based method whose predictions depend entirely on the ranking of gene pairs in each sample[98,99]. Based on the average logFC, we divided the markers of the PRN clusters into positive (average logFC > 0) and negative (average logFC < 0) markers. We then paired the top positive

and negative markers (100 genes each) together to build a biological mechanism representing the PRN mesenchyme (30,000 pairs). Each pair consists of two genes, one is up- and another is down-regulated in the PRN mesenchyme. This mechanism was then used as a priori biological constraint during the training of the k-TSPs algorithm[100], and the resulting signature was evaluated on the indepedent testing set.

In addition, we evaluated the prognostic relevance of this signature in the TCGA cohort which included 493 primary PCa samples. First, we built a logistic regression model using the 26 genes comprising the PRN signature and used this model to generate a probability score for progression-free survival (PFS) in each patient. We then binarized these probabilities into predicted classes and compared their PFS probability using Kaplan–Meier survival analysis[101]. Finally, we calculated the hazard ratio (HR) of the signature prediction probability scores after adjusting for Gleason grade using a multivariate Cox proportional hazards (CPH) model[102].

### Comparing the predictive and prognostic performance of the stromal PRN and cell-cycle progression signatures

We retrieved a cell-cycle progression (CCP) signature consisting of 31 genes[67] and used it to develop a predictive model for metastasis. Both the PRN and CCP signatures were trained and tested on the same training and testing sets described above, utilizing a logistic regression model to predict metastatic events. The training and testing performance of both signatures was compared using the Area Under the Reciever Operating Characteristics Curve (AUROC). Implementing the same approach used for the PRN signature, we tested the association of the CCP signature with PFS in the TCGA cohort using Kaplan–Meier survival analysis and a multivariate CPH model adjusting for Gleason grade.

### Histopathology studies

Following radical prostatectomy, human prostates were submitted for gross pathological assessment and sectioning, with ischemic time less than 1 h. The prostate specimen was serially sectioned from apex to base into 3–5 mm slices. In prostates with grossly identifiable tumor, a 5 mm biopsy punch was taken from the area of tumor, an area adjacent to the tumor, and an area distant (>2 slices away) from the tumor. In prostatectomy specimens where tumor was not definitively grossly visible, these areas were approximated by anatomic correlation of the MRI findings and targeted biopsies with the highest tumor grade (as described in ref. 70).

The prostate slices were fixed in 10% buffered formalin, embedded in paraffin blocks, and hematoxylin & eosin (H&E)-stained slides were created, per routine clinical pathologic assessment. Upon evaluation of the H&E slides, a urologic pathologist (BDR) confirmed that the punched area of tumor, area adjacent to tumor, and area distant to tumor were accurately represented based on the histology of the areas surrounding the punched area. Prostate from WT and GEMM mice were dissected. One-half of the prostate from GEMMs was utilized for scRNA-seq (see above). The contralateral half was fixed in 10% buffered formalin and embedded in paraffin blocks, sections were cut, and hematoxylin & eosin (H&E)-stained slides[103,104]. Collagen deposition in the different GEMMs was assessed by Masson's trichome staining[105,106], followed by collagen deposition quantification digitally performed using HALO (Indica Labs, v3.3.2541, Albuquerque, US). A HALO-based digital classifier was developed to identify collagen, epithelium, muscle fiber, and background regions on the digital images. Percentages of collagen deposition were then quantified and compared using unpaired t-test. Immunohistochemical stainings were used to confirm the expression of the GEMMs proteins. Primary antibodies used for IHC staining were: Rabbit monoclonal Recombinant Anti-c-Myc antibody [Y69] (Abcam # ab32072; 1:100); rabbit monoclonal PTEN (D4.3) XP (Cell Signaling #9188S; 1:125); rabbit monoclonal P-AKt (S473) (D9E) XP® (Cell Signaling #4060S; 1:100); rabbit monoclonal N-Myc (D4B2Y)

(Cell Signaling #51705S; 1:100); rabbit monoclonal Anti-ERG antibody [EPR3864] (Abcam #ab92513; 1:1000). Secondary antibodies used in IHC were the Poly-HRP IgG reagent from the BOND Polymer Refine Detection Kit (cat DS9800, Leica Biosystems). Immunohistochemistry to interrogate for panel markers (Supplementary Data file 7) was performed on 5-μm-thick formalin-fixed paraffin-embedded tissue (FFPE) of (i) human PCa and (ii) GEMMs sections using previously-established protocols[103,107,108].

Multiplexed immunohistochemistry (mIHC) was performed by staining 5-μm-thick FFPE core biopsy sections in a BondRX automated stainer, using published protocols[109–111]. One panel of primary antibody/fluorophore pairs was applied to all cases along with Antibody/ Akoya Opal Polaris 7-Color Automated IHC Detection Kit (NEL871001KT), and Opal Polymer Anti-Rabbit HRP kit for secondary antibody (ARR1001KT) fluor combinations were utilized as follows (Supplementary Data file 7). Primary antibodies used for immunofluorescence were: rabbit monoclonal anti-Gpx3 [EPR22815-112] (Abcam; #ab256470; 1:200); rabbit monoclonal anti-C3 [EPR19394] (Abcam; # ab200999; 1:10,000); rabbit monoclonal anti-Wif1 [EPR9385] (Abcam # ab155101; 1:5000); Purified Mouse Anti-Synaptophysin [2/synaptophysin (RUO)] (BD Biosciences #611880; 1:100); rabbit monoclonal Anti-Periostin antibody [EPR20806] (Abcam # ab215199; 1:1000); rabbit monoclonal Recombinant Anti-Androgen Receptor antibody EPR179(2) (Abcam #ab108341; 1:1000); rabbit polyclonal Anti-pan Cytokeratin antibody (Abcam #ab217916; 1:400). Secondary antibodies used for immunofluorescence were: the anti-rabbit Akoya Rabbit HRP (cat #ARR1001KT, Akoya Biosciences) and the anti-mouse Mouse Superboost (cat #B40961, Thermo Fisher Scientific). The tyramide-conjugated fluorophores were Opal 480 (cat #FP1500001KT, Akoya Biosciences; 1;75); Opal 520 (cat #FP1487001KT, Akoya Biosciences; 1:75); Opal 570 (cat #FP1488001KT, Akoya Biosciences; 1:100); Opal 690 (cat #FP1497001KT, Akoya Biosciences; 1:100); Opal 780 (cat #FP1501001KT, Akoya Biosciences, Opal 780 dilution 1:15, TSA-DIG dilution 1:50).

The order of processing slides was as follows: primary antibody incubated for 30 min; Blocking for 5 min with Akoya Blocking/Ab Diluent; Opal Polymer Anti-Rabbit HRP incubated for 30 min; Opal 480-690 incubated for 10 min; Leica Bond ER1 solution incubated for 20 min. All slides were also stained with DAPI for nuclear identification.

### Acquisition and computational analysis of multiplexed immunofluorescence images

Whole slide images of hematoxylin and eosin, trichrome, and mIHC sections were acquired using the Vectra Polaris Automated Quantitative Pathology Imaging System (Akoya Biosciences, Hopkinton, MA)[112]. Images were processed by linear spectral unmixing and deconvolved[113]. Cells were segmented and a human-in-the-loop HALO random forest (RF) classifier was trained with labels from a pathologist to select stromal cells. Subsequently, these stromal regions of the entire prostate surrounding glands in WT and GEMMs mice were preprocessed and analyzed using PathML (v2.0.0) (https://github.com/Dana-Farber-AIOS/pathml)[114] to generate a single cell counts matrix containing statistics summarizing the expression of each protein in each cell together with the cell size, coordinates, and eccentricity. To address technical artifacts in the segmentation results, DAPI-negative cells were filtered out. In addition to the HALO RF classifier, we used PanCK to validate the presence of epithelial (PanCK+) and stromal cells (PanCK-), respectively. A nearest-neighbor graph was constructed from the counts matrix using Euclidean metric as implemented in the Scanpy package[36]. This graph was clustered using the Leiden algorithm[79] to identify subpopulations of cells and low-quality cells. Cells were projected to two dimensions and visualized using the UMAP algorithm[82]. A binary label indicating the presence/absence of

each protein was created by thresholding markers for positive or negative signal with pathologist assistance.

## Spatial transcriptomics analysis

We conducted spatial transcriptomics analysis of murine PRN tumors and corresponding tissue from its wild type using 10X Genomics CytAssist Visium platform (10x Genomics, Pleasanton, CA). Prostate from WT and GEMM mice were dissected. One-half of the prostate from GEMMs was utilized for scRNA-seq (see above). The contralateral half was fixed in 10% buffered formalin and embedded in paraffin blocks and sliced into sections with a thickness of 10 μm thickness. dried at 42 °C for 3 h and kept in a desiccator at room temperature overnight, before proceeding with the Visium CytAssist spatial protocol (guides CG000518, CG000520, and CG000495). Slides were deparaffinized, H&E stained, and imaged using an EVOS M7000 Automated Imaging System (10x objective, 3.45 μm/pixel - Thermo Fisher Scientific, CA). Slides were then de-coverslipped and tissues were hematoxylin destained, decrosslinked and hybridized overnight with the whole mouse transcriptome panel which contains pairs of specific probes for each targeted gene (PN-1000365). After hybridization, the probe pairs were ligated, the slides loaded on a Visium CytAssist instrument, ROIs adjusted and ligated probes transferred and captured on an 11 mm Visium CytAssist Spatial Gene Expression slide containing UMIs and barcoded oligos. Spatially barcoded libraries were generated and sequenced with paired-end dual-indexing (28 cycles Read 1, 10 cycles i7, 10 cycles i5, 90 cycles Read 2) Sequencing libraries were demultiplexed with bcl2fastq (Illumina). Spatial transcriptomics libraries were processed and aligned to the mm10 mouse reference genome using the Space Ranger software (version 2.0.1), and tissue-associated barcodes were kept for further downstream analysis. The filtered UMI count matrices were merged to enable their joint analysis. Subsequently, the data was normalized (each cell was normalized by total counts over all genes) and log-scaled, then the top 4000 highly variable genes were identified (by model). The neighborhood graph was computed using the first 10 prinicipal components and 15 neighboring data points, then embedded using the UMAP algorithm. Spots were clustered using the Leiden algorithm with a resolution of 0.5, then marker genes were computed by comparing each cluster to the remaining clusters using a Student's $t$-test. Spot annotation was performed using the clusters marker genes.

## Generation of murine normal associated fibroblasts (NAFs)

Prostate tissues derived from 3-month-old C57/BL6 male mice were minced in apron 1 mm pieces and placed in p100 using DMEN + 5% FBS + 5%NuSerum + 1%Gln + 1%P/S + 10nMDHT. The fibroblasts were attached to the plate within 48–96 h and the chunks were removed. Then the immortalization was performed using Retrovirus with zeocin resistance and expression of SV40 T antigen (pBabe-Zeo-LT-ST). NAFs were cultured in normal DMEM + 10%FBS + 1%Gln+1%P/S.

## RNA knockdown

For lentiviral shRNA transduction, mouse NAFs were transduced using lentiviruses containing shRNA constructs against *Postn* with 10 mg/ml polybrene (Sigma, TR-1003-G). shPostn1 (F primer: CACCGGGCCAT TCACATATTCCGAGAACTCGAGTTCTCGGAATATGTGAATGGCTTTTT G; R primer: GATCCAAAAAGCCATTCACATATTCCGAGAACTCGAGT TCTCGGAATATGTGAATGGCCC).

shPostn2 (F primer: CACCGGCCACATGGTTAATAAGAGAATCTC GAGATTCTCTTATTAACCATGTGGTTTTTG; R primer: GATCCAAAAA CCACATGGTTAATAAGAGAATCTCGAGATTCTCTTATTAACCATGTGG CC) and shCtrl: Fw: CACCGGCCTAAGGTTAAGTCGCCCTCGCTCGAG CGAGGGCGACTTAACCTTAGTTTTTTG.

Rv: GATCCAAAAAACTAAGGTTAAGTCGCCCTCGCTCGAGCGAG GGCGACTTAACCTTAGGCC.

## Migration (invasion) assay

For Boyden chamber assays, 100,000 NAFs infected with control of Periostin-directed shRNAs (shCtrl, shPostn1, or shPostn2) were seeded into a 24-well plate in culture media for 24 h. Cells were washed twice for 15 min in minimal media (DMEM (Thermo Fisher, 31053036) with 1× penicillin/streptomycin (Gibco, 15140-122), 1× GlutaMAX (Gibco, 35050-061) and 10 mM HEPES (Gibco, 15630-130). Cell culture inserts (Millipore, #MCEP24H48) were coated with Matrigel (Corning, 354230) diluted 1:10 in PBS, and incubated for 2 h +37 °C. 22rv1 shRb1 NMYC cells were harvested, washed twice for 3 min in minimal media, and seeded in triplicate at a density of 75,000 cells/ insert in 200 μl. Inserts were placed into empty 24-well plates and incubated for 15 min at +37 °C and 5% $CO_2$ before transferring into the test conditions, minimal media was used as a negative control and minimal media supplemented with 10% charcoal-stripped serum (Gibco, A33821-01) was used as a positive control. Cells were then allowed to migrate at +37 °C for 6 h. Filters were fixed using 4% PFA/ PBS, washed with PBS, and stained using Hoechst before washing, cleaning, and mounting using Fluoromount-G (SouthernBiotech, 0100-01). Cells that had migrated through the filter were quantified (5 fields of view per filter) and normalized to the negative control. 22rv1 shRb1 NMYC cell line was kindly provided by Dr. David S. Rickman. Both human and murine cell lines routinely tested negative for the presence of mycoplasma, which was performed using a mycoplasma detection kit (abm #G238).

## Co-culture of fibroblasts and epithelial cells derived from GEMMs

For co-culture experiments, primary prostate fibroblasts were derived from 12-week-old FVBN mice (JAX). Epithelial cells were derived from the T-ERG and PRN mouse models. On day 1, individual 12 mm Transwell® with 0.4 μm pore polyester membrane inserts (Corning #3460) were coated with a 100 μg/ml solution purified bovine type I collagen (PureCol®, Advanced BioMatrix #5005) on both membrane sides, following the manufacturer's protocol. After that, 50,000 FVBN fibroblasts were seeded on the lower side of the membrane in a 150 μl drop of complete medium (DMEM + 10%FBS + 1%Gln+1%P/S), left attached to the membrane with the insert upside down for about 6 h, and then placed back in a well of a 12-well plate with complete medium. The day after, the Transwell® inserts were transferred into fresh plates with complete mouse organoids medium[115] supplemented with 3% Fetal Bovine Serum (FBS). Organoids cells were detached and single-cell suspended using TrypLE, pelleted and counted. Then, either 50,000 T-ERG or PRN epithelial organoid cells were resuspended in 200 μl of complete mouse organoids medium + 3% FBS and seeded on top of the Transwell® membrane. The cells were left in co-culture, or as FVBN fibroblasts-only as controls, up to day 9 (8 days of co-culture), changing medium in the wells every 2 days. Next, control and epithelial-induced fibroblasts were collected and submitted to 10x single cell RNAseq.

## Statistics and reproducibility

For differential gene expression testing, a two-part generalized linear model, known as the hurdle model, was utilized through the MAST framework. Gene regulatory network activities were inferred from the raw counts matrix with the SCENIC pipeline. Clustering and data visualization were achieved using algorithms such as the Leiden clustering algorithm, Partition-Based Graph Abstraction (PAGA), and Uniform Manifold Approximation and Projection (UMAP). Label transfer from the mouse to human scRNA-seq was carried out via the 'ingest' method. For the development of the PRN signature to predict metastasis, stratified sampling was implemented to ensure balanced representation across the training and testing cohorts, and the k-top scoring pairs (k-TSPs) algorithm was used for classifier training. Survival probabilities were estimated using the Kaplan–Meier method and

evaluated by the Log-rank test. Multivariate survival analysis was performed using the Cox proportional hazards (CPH) model and was evaluated using the Wald test.

## Reporting summary

Further information on research design is available in the Nature Portfolio Reporting Summary linked to this article.

## Data availability

The single-cell RNA-seq and Visium Spatial Transcriptomics data generated in this study has been deposited in the Gene Expression Omnibus (GEO) under the accession codes: GSE244267, GSE244269, and GSE248466. The gene expression publicly available data used in this study are available in GEO under the accession codes GSE116918[88], GSE55935[89], GSE51066[90], GSE46691[91–93,116], GSE41408[94], and GSE70769[95]. The processed count matrices for the single-cell RNA-seq, Visium spatial transcriptomics data, together with the expression matrix and phenotype labels of the natural history cohort have been deposited in Zenodo https://doi.org/10.5281/zenodo.7452769[97]. The microscopy data reported in this paper will be shared by the lead contact. The remaining data are available within the Article, Supplementary Information or Source data file. Source data are provided with this paper.

## Code availability

All original code can be accessed through the GitHub public repository (https://github.com/MohamedOmar2020/pca_TME) and https://doi.org/10.5281/zenodo.8357518[117].

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

## Acknowledgements

We thank Dr. Owen Witte for critical review of the manuscript and valuable suggestions; and Dr. David Goodrich for providing histology slides of DKO and TKO GEMMs for analysis. We are grateful to the fol-lowing facilities at Weill Cornell Medicine for their contributions to this work: the Multiparametric In Situ Imaging (MISI) Laboratory, the Geno-mics, the Epigenomics, and Flow Cytometry cores; the Research Animal Resource Center, and the Pathology Clinical Genomics Lab. This work was supported in part by National Cancer Institute grants: prostate cancer SPORE P50CA211024, P01 CA265768, R01s: CA200859, CA173481 and CA183929, T32CA260293. We acknowledge support from the Prostate Cancer Foundation 2022CHAL05, the DoD W81XWH-19-1-0566, the Pan Prostate Cancer Group (PPCG), the National Science Foundation grant 2124167, and the "Ezio, Maria e Bianca Pan-ciera" AIRC Fellowship for Abroad.

## Author contributions

H.P., R.C., M.O., and M.L. conceived and designed the study. H.P., M.J., F.P., C.S., L.V.E., S.R., C.F.-R., N.J.B., C.U., M.K.A., and F.N.-d.-A. collected mouse and human data. C.E.B., T.P., F.K., M.K.A., and B.R. provided study materials or patient samples and H.P., R.C., M.O., G.N.F., F.P. T.P., L.V.E., N.J.B., R.G., C.A.S., M.B.G., F.K., B.R., P.V.N., D.S.R., M.L., and L.M. investigated, analyzed and interpret data, R.C., M.O., W.D., and L.M. processed and analyzed single-cell and spatial transcriptomics data, T.P., F.S., G.N.F., I.V., C.U. C.S., and J.S. performed immunohistochem-istry. H.P., R.C., F.P., M.O., L.M., C.E.B., D.S.R., and M.L. wrote and reviewed manuscript. All authors edited and approved the manuscript.

## Competing interests

The authors declare no competing interests.
