## [Peer Review File · Nature Communications]

Distinct mesenchymal cell states mediate prostate cancer progressionREVIEWER COMMENTS

Reviewer #1 (Remarks to the Author):

This study presents the largest characterization of stromal cells in prostate and prostate cancer models, yielding a new perspective on stromal heterogeneity. The computational analyses nominate many potentially important signaling networks and transcriptional regulators mediating stromal-epithelial or stromal-immune crosstalk. This collaborative effort across labs and mouse models is impressive and provides a useful resource for the field. I especially appreciated the validation that different stromal cell subtypes are found across species and that ERG-enriched and NEPC-enriched stromal signatures are conserved in human prostate. Given that the larger perspective of the study comes from the concept that epithelial-specific genomic changes (ERG, NEPC, etc.) reprogram the stromal microenvironment, there is a notable absence of functional experiments to verify the relationship in this direction. Something to prove this point would strengthen the study considerably. Additional comments are below:

1. Figure 1E contains a lot of information but just a few sentences in the text, and then this sort of analysis is presented throughout the manuscript. It would be better for readers if the authors could go into a bit more depth here of what is being evaluated and what was found. For example, what does the collagen signaling network mean exactly here? Is there a gene signature of collagen activation? Is this literally referring to collagen genes from the RNAseq? Additionally, what do the authors make of the fact that no signals appear to be going from stroma to epithelium in the wild-type prostates?
2. In Figure 3, the authors look at C1 markers GPX3 and C3 in the tissue but these appear to be expressed at similar levels in c5, so this should be discussed. If there isn't a marker that is entirely specific for this subtype, please make that clear to readers.
3. With the ligand-receptor interaction analysis in Figures 3 and 4, perhaps it is not surprising that the same signaling networks from stromal cells signal to epithelium and immune cells. But could it be coming from a bias in the way the data is analyzed? Are there simply more representative genes/pathways related to collagen in stromal cells and this is why it comes up as the most significant in each of these plots? This is not my area of expertise but I am sure some readers will have the same questions.
4. The idea that stromal cells reflect the AR activity of the nearby tumor cells (AR+ stroma in AR+ cancer models, AR- stroma in NEPC models) is quite interesting. It would be great to verify the abundance of POSTN+ vs AR+ stromal cells in additional GEM models of NEPC using IF/IHC.
5. The K-M plot in Figure 5G is impressive but I wonder how much of this is truly due to stromal cells in TCGA tissues used for sequencing and how much of it is due to the proliferative signature of these stromal cells. Can the authors computationally remove the proliferation genes and see if the more stromal-specific genes in the signature are driving the outcome (progression-free survival)?
6. In Figure 6B, it appears as if most of these markers are expressed similarly across all stromal subtypes in contrast to data shown in mouse models in Figures 3A and 4A. Do the authors agree, and is there a hypothesis why this is the case?

7. An important take-home message from this study is that different alterations in epithelial cells can reprogram the stroma. But most of the analysis in this paper is in the opposite direction, how the stromal cells signal to the epithelium. The only functional experiment in the paper is also about how the stromal cells regulate invasiveness of tumor cells. Can the authors include a functional experiment to show how epithelial cells reprogram the stroma, which would be aligned with their stromal heterogeneity data?

8. Several typos are noted. For example “of of” in the abstract and “stroma-epithelial interactions” on line 75 and “high frequency” on line 234. Please do a complete read-through for small edits and spelling before submitting the revised manuscript.

Reviewer #2 (Remarks to the Author):

SUMMARY

The manuscript "Distinct mesenchymal cell states mediate prostate cancer progression" (Pakula et al.) identifies and describes common and distinct stromal populations as well as GEMM-specific transcriptional programs, using three murine model systems. The authors focus on the mesenchymal (stromal) cells and their role in prostate cancer progression, analyzing 8,574 mesenchymal cells from a total of ~101,000 cells. They also look at similarities between mouse and human samples. They also identify a gene signature from PRN/NEPC-associated mesenchymal cell clusters (C5-C7), that may predict metastatic progression and PFS independent of Gleason score.

GENERAL COMMENTS

- Overall, I quite liked this manuscript. It is clear and well-written. The figures are nicely organized.
- The findings will help us understand the contribution of stromal cells in prostate cancer progression and metastasis and identify subpopulations of cells that may allow for therapeutic targeting.
- The manuscript represents a valuable resource for the field. The manuscript is almost entirely descriptive and almost entirely based on transcriptomic data. This is not a criticism, but an observation. This could be published as a resource, and that resource would be useful for the research community.
- It would elevate the importance of the findings if the authors were able to validate key findings, e.g., by spatial transcriptomics, multiplex immunohistochemistry. As it stands, the biologic relevance of the author's findings is interesting but unproven.
- The validation data in Figure S6 (periostin and androgen receptor expression) is an attempt at validation, but is not very convincing in the current presentation and resolution, and lacks other stromal and tumor markers to confirm the spatial location where periostin and androgen receptor are being expressed.

SPECIFIC COMMENTS

- The analysis of cell-cell communications is a little superficial. Besides the general signaling pathways, did the authors observe specific ligand and receptor interaction channels across the different stromal cell subsets? These specific channels could be listed. Are the cell communications pathways different between the three mouse models. Does the human data

show similar interaction patterns?

- Epithelial cells are heterogeneous, including different subsets such as luminal, basal, hillock and club. Is there a population of epithelial cells that more closely interacted with stroma cells? Would it be useful to include tumor cells into the analysis?
- Results, line 123, “an additional cluster was removed as it had <5% of cells in WT mice”. This immediately sounds like a tumor specific mesenchymal cell population that would be of particular interest. Is this the case, or do the authors simply mean that there were too few cells for a meaningful statistical analysis?
- Figure 1A and 1E. I do not like the term “mutant(s)”, when I believe you are referring to the different GEMMs. Can this be labeled GEMMs for example?
- Figure 1B and 1C could be moved to supplemental if space was needed.
- Results, line 141. Figure S2 and Table S3. I like the idea of the regulons and the differential gene expression, though perhaps the authors could add 1-2 sentences to expand upon (a) how do the authors define a regulon, and (b) what criteria / cutoff identifies a differentially expressed gene in one cluster as compared to the other clusters? The authors use the regulon data quite a lot in subsequent results, and so it would be worth the time to expand on the explanation.
- Results, line 250. “In order to assess whether Postn-positive stroma facilitates invasion, a characteristic of NEPC, a migration assay was utilized.”. This is an intriguing finding, but I think it is an over-interpretation to say that stromal motility facilitates tumor cell invasion. Given that the migration is affected by POSTN knock-down, it would be important to show that the cells remained viable and otherwise healthy, confirming that the decreased migration was not simply due to unhealthy cells.
- Results, line 262. “The transcriptional profiles of the PRN-derived clusters are predictive of metastatic progression in prostate cancer”. Here the authors took gene pairs expressed in a subset of the ~9000 mesenchymal cells (only clusters c5-c7). Then they used this signature to probe large bulk-RNA datasets, finding quite a nice AUC and quite a striking PFS difference. I worry about the significance of this finding, given that c5-c7 cells represent a maximum of 5% of the cells in a tumor (i.e., ~5000 cells out of ~100K cells that the authors sequenced) so it makes me wonder whether it is the expression within the mesenchymal cells driving these differences. Are those gene pairs expressed in other cells (i.e., tumor cells, immune cells, epithelial cells, etc.?) and is that possibly driving the findings or are these genes truly specific to the mesenchymal clusters c5-c7 or are they seen in other cell types within the tumor? The authors do have the data on the other 95,000 cells.
- Cell type specific marker gene expression is not clear for c0 to c8. Many markers are commonly expressed in multiple clusters. (Figure 3A, 4A, 6B).
- It would be beneficial to create two gene expression plots which compare the mouse and human data projection with matched marker genes.
- Did you check NEPC single cell data to see if the three PRN/NEPC-(c5-c7) that were less abundant in human tumors are NEPC specific stromal cells? There exist public NEPC single cell datasets.
- Did you observe any batch effect when combining bulk RNAseq datasets?
- In Figure 4A, it seems that Wnt pathway related genes also show expression in C0 and c1. The DotPlot is not ideal for demonstrating the differences, we would recommend a violin type of plot with statistics.
- Statistical significance for Figure 6D is missing.
- Although the selected each GEMMs represent different pathological changes and disease stages, it would be great if the authors provided detailed information about the timing of harvesting the samples.

Reviewer #1

Comment 1: This study presents the largest characterization of stromal cells in prostate and prostate cancer models, yielding a new perspective on stromal heterogeneity. The computational analyses nominate many potentially important signaling networks and transcriptional regulators mediating stromal-epithelial or stromal-immune crosstalk. This collaborative effort across labs and mouse models is impressive and provides a useful resource for the field. I especially appreciated the validation that different stromal cell subtypes are found across species and that ERG-enriched and NEPC-enriched stromal signatures are conserved in human prostate. Given that the larger perspective of the study comes from the concept that epithelial-specific genomic changes (ERG, NEPC, etc.) reprogram the stromal microenvironment, there is a notable absence of functional experiments to verify the relationship in this direction. Something to prove this point would strengthen the study considerably.

Comment 2: An important take-home message from this study is that different alterations in epithelial cells can reprogram the stroma. But most of the analysis in this paper is in the opposite direction, how the stromal cells signal to the epithelium. The only functional experiment in the paper is also about how the stromal cells regulate invasiveness of tumor cells. Can the authors include a functional experiment to show how epithelial cells reprogram the stroma, which would be aligned with their stromal heterogeneity data?

Response to Comments 1 and 2: We thank the reviewer for the positive comments and for the important points raised. To address these, we have re-performed the ligand-receptor interaction analysis to infer the signaling received by the stromal clusters from the epithelial as well as the immune compartments. The data support the claim that different epithelial mutations shape the stromal microenvironment in unique ways.

A. UPDATED BIOINFORMATIC ANALYSIS

The **GEMMs versus WT** section in Results now includes: *“Using ligand-receptor (L-R) interaction analysis, we compared both the number and strength of signaling interactions between the stroma, epithelium, and immune compartments in both WTs and GEMMs where both were significantly higher (**Figure 1C**). Specifically, outgoing signaling from the stroma of GEMMs is mainly mediated through interactions between subunits of collagen type I, III and IV on the stroma and their corresponding receptors, the collagen-binding integrin *Itga2/Itgb1* on the epithelium (**Table S3**). Additionally, epithelium signaling to the stroma is mediated through the *Wnt4-Fzd1/Fzd2* and *Areg-Egfr* interactions (see **Figure 1D** and **Table S3**). On the other hand, the only significant epithelial-stromal interaction in WTs is between the ligand *Gas6* on the epithelium and its receptor *Axl* in the stroma (**Figure 1D**). Similarly, we found that signaling from immune to stromal cells is significantly increased in GEMMs compared to WTs, mainly through the *Hbegf-Egfr* and *Mif-Ackr3* interactions (see **Figure 1D** and **Table S3**). Overall, these results show that specific epithelial mutations do not just alter the stromal composition, but also induce significant changes in the inter-cellular communication networks in the microenvironment.*

The **common clusters (c0-c2)** results section in Results now includes this paragraph: *“L-R interaction analysis revealed several communication networks from the basal and luminal epithelium to c0 and c1 stromal cells (**Figure 3C**). These communications are mostly mediated through interactions between *Thbs1* found on luminal cells and *Sdc1*, *Sdc4*, and *Cd47* found on the stromal cells of c0 and c1, and also through interactions between *Mif* expressed in luminal cells and *Ackr3* expressed mainly in c1 (see **Figure 3C** and **Table S3**). Notably, we found a distinct distribution of immune cell types across different mouse models (**Figure 3D**). Immune signaling to c0 and c1 is dominated mainly by macrophages and to a lesser extent by dendritic cells (**Figure 3E**). Macrophages signaling to c0 and c1 is mediated mainly through *Spp1**

(expressed in macrophages) and integrins (expressed in c0 and c1). Specific Macrophages-c0 interactions are conducted through *Gzma* on macrophages and *Pard3* on c0, in contrast to specific macrophages-c1 interactions, which are mediated through the ligands *Mif* and *Hbegf* on macrophages and their corresponding receptors *Ackr3* and *Egfr* on c1 cells.”

For the T-ERG, Pten, and Hi-MYC-specific clusters (c3-c4), the result section now includes the following paragraph: “... in T-ERG, signaling from luminal cells to c3 and c4 stroma are mainly mediated through *Edn1-Ednrb*, *Thbs1-Sdc1/Sdc4*, and *Wnt4-Fzd1/Fzd2* interactions, while basal-stromal signaling is mainly mediated through *Gas6-Axl*, *Col4a1-Sdc4*, and *Jag1-Notch2* interactions (see **Figure 4C** and **Table S3**). On the other hand, luminal-stromal signaling in Hi-MYC is mediated solely by the *Mif-Ackr3* interaction, while basal-stromal signaling is conducted mainly by interactions between *Tgfb1* and its receptor *TGFbR1* (**Table S3**). Finally, in the NP mouse model, there is an increased activity of *Wnt* signaling from luminal and basal cells to c4 stroma, mainly through interactions between *Wnt4* and *Wnt7b* and their receptors on stromal cells including *Fzd2* and *Fzd5* (**Table S3**). Immune-mediated signaling to c3 and c4 stroma also shows significant differences between the T-ERG, Hi-MYC, and NP models. For instance, in T-ERG, signaling from NK and cytotoxic T cells to the c3-c4 stroma are mediated mostly through *Fas-Fas* and *Gzma-F2r* interaction (**Figure 4C**) while in Hi-MYC, these networks involve mainly the *Lgals9-Cd44* interaction (**Table S3**). Similarly, signaling from dendritic cells to c3-c4 stroma was mediated through different L-R interactions across the three mouse models, With T-ERG characterized by increased activity of *Wnt11-Fzd1* and *Nectin1-Nectin3* interactions (**Figure 4C**), while Hi-MYC and NP were characterized by increased activity of the *Spp1-Itgav/Itga5* interaction.”

For the NEPC-specific clusters (c5-c7), the Results section now includes the following paragraph: “... signaling from the luminal and basal epithelium to the stroma is mediated mainly through the *Tgfb1-TgfbR1* and *Tnf- Tnfrsf1a* interactions, as well as WNT-mediated signaling involving interactions between *Wnt4*, *Wnt7b*, and *Wnt10a* and the WNT receptors *Fzd1*, *Fzd2*, and *Fzd3*. In the immune TME, the inferred macrophages signaling to the PRN mesenchyme is driven mainly by interactions between *Spp1* and *Fn1* (expressed in macrophages) and their receptors on c5-c7 including *Sdc1*, *Sdc4*, *Itga5*, *Itgb5*, *Itgav*, and *Cd44* (**Figure 4D**). Several L-R interactions are also inferred between the interleukins *Il1a* and *Il1b* on macrophages and their receptor *Il1r1/Il1rap* (c5), and between *Tnf* and its receptors *Tnfrsf1a* and *Tnfrsf1b* (c5-c7). Unlike other clusters, c5 is also characterized by receiving signals from regulatory T cells (Tregs) through interactions between *Il17a* expressed in Tregs and its receptors *Il17ra* and *Il17rc* expressed solely on c5 (**Figure 4D**). Importantly, we inferred several communication networks between NK/cytotoxic T cells and the PRN mesenchyme which are driven by the Interferon-gamma (IFN γ) signaling pathway (**Figure 4D**). These networks are also observed between NK/cytotoxic T cells (expressing *Ifng*) and the tumor epithelium, (expressing *Ifngr1* and *Ifngr2*). Stromal signaling through the *Periostin* pathway in particular is restricted to the PRN mesenchyme with few interactions involving c0 and c1 and no statistically significant interactions involving c3 and c4 (**Figure S3B** and **Table S3**).”

B. FUNCTIONAL VALIDATION

To provide a functional validation of at least part of our observations, we performed an *in vitro* co-culture assay to confirm that specific epithelial genotypes determine specific mesenchymal phenotypes. To this end, we co-cultured primary fibroblasts derived from FVB/N male mice alone, in combination with T-ERG-derived organoids, or in combination with PRN-derived organoids (**Figure 5G**). Epithelial cells are indeed able to shift the phenotype of co-cultured fibroblasts

towards the most representative ones for each genotype (i.e., c3 for T-ERG, c6/c7 for PRN) in scRNA-seq space, thus strengthening and validating our observations.

To this end, we added the following paragraph to the Results section: "...specific epithelial mutations can shape the stromal microenvironment in PCa, as suggested by the distinct expression profiles of T-ERG, NP, and Hi-MYC-specific stromal clusters (c3-c4) compared to the PRN-derived clusters (c5-c7). To functionally validate these findings, we co-cultured normal fibroblasts from the FVBN mouse model with epithelial cells from the T-ERG and PRN models. We found that fibroblasts co-cultured with epithelial cells from the T-ERG model tended to exhibit similar expression profiles to c3 and c4 stromal cells, while those co-cultured with PRN epithelium exhibited expression profiles of the c5-c7 stromal clusters with high Postn expression (Figure 5G)."

Figure 1E contains a lot of information but just a few sentences in the text, and then this sort of analysis is presented throughout the manuscript. It would be better for readers if the authors could go into a bit more depth here of what is being evaluated and what was found. For example, what does the collagen signaling network mean exactly here? Is there a gene signature of collagen activation? Is this literally referring to collagen genes from the RNAseq? Additionally, what do the authors make of the fact that no signals appear to be going from stroma to epithelium in the wild-type prostates?

Response: We thank the reviewer for raising this important point. **Figure 1E (Figure 1C-D in the revised manuscript)** highlights the differences in signaling networks between the stroma, epithelium, and immune compartments in GEMMs compared to wild types. Specifically, the heatmap in **Figure 1C** shows the number and strength of outgoing signaling from each compartment (columns) and each pathway (rows), which demonstrates a significant increase in the overall number of signals in GEMMs compared to wild types (see bar plot on top). The chord diagrams in **Figure 1D** show the specific pathways involved in mediating the signaling between these three compartments in GEMMs and wild types.

We did not observe statistically significant signaling networks from the stroma to the epithelium in the prostates of WTs which we explained in the revised manuscript using the following paragraph in the Results section (page 7): "..... we did not observe any statistically significant signaling networks from the stroma to the epithelium in the prostates of WTs since these did not pass the filtering thresholds that we implemented to eliminate false positive results. This suggests that while stroma to epithelium interactions do exist in normal prostates, they occur at a lesser frequency and strength compared to those found in GEMMs."

We revised the figure legend and the Results section accordingly and added a new supplementary table (**Table S3**) which offers a comprehensive view of these interactions including the individual ligands and receptors, corresponding pathway, interaction probability, and the expression levels of ligands and receptors on the source and target cells, respectively.

In Figure 3, the authors look at C1 markers GPX3 and C3 in the tissue but these appear to be expressed at similar levels in c5, so this should be discussed. If there isn't a marker that is entirely specific for this subtype, please make that clear to readers.

Response: The reviewer is correct in that Gpx3 and C3 are expressed in both c1 and c5 albeit at different levels. We concur therefore that if these markers are used to exemplify c1 cluster specificity, this is misleading. We therefore provide a more generic statement in the results section

(page 9) which now reads: “The common cluster c1 shows increased expression of *Sfrp1* and *Gpx3* and of major complement system components such as *C3*, *C7*, and *Cfh* compared to wild type. ... Enrichment of *Gpx3* and *C3* in the stroma surrounding PIN and invasive tumor can be seen in **Figure 3B**”.

In reality, at the mRNA level, there is a rank order of markers from highest to lowest expression that specify the different clusters (see **Table S4** now added to the revised manuscript). The

Figure R1. Representative IHC images for Wif1 staining in WT and T-ERG. All images are at x40 magnification with a scale bar of 20 μ m.

specificity of the top-ranked genes, however, is evident only in the GEMM-specific clusters (c3-c7). To show this, we added a panel in **Figure 4** in which we demonstrate the expression of a gene, Wif1 (see **Table S4**, rank order number 10 for c3), that shows specificity for the TRG model, with little to know

expression in the other GEMMs and only a restricted expression in the myofibroblasts in the wild type (**Figure R1**).

With the ligand-receptor interaction analysis in Figures 3 and 4, perhaps it is not surprising that the same signaling networks from stromal cells signal to epithelium and immune cells. But could it be coming from a bias in the way the data is analyzed? Are there simply more representative genes/pathways related to collagen in stromal cells and this is why it comes up as the most significant in each of these plots? This is not my area of expertise but I am sure some readers will have the same questions.

Response: This is a very important point. The probability and significance of different pathways are not confounded by their number of genes (ligands and receptors) since they were computed for each ligand and receptor pair independently, and were subsequently aggregated at the pathway-level (see **Methods section, page 42**) using the methods described¹.

The revised Results section and Table S3 show that the collagen network, mediating signaling from the stroma to epithelial and immune cells. This gene set is not more extensive than others but it is utilized to an extent far and above other pathways to communicate with epithelial cells in particular. It includes a family of genes coding for several subtypes of collagen associated with (a) ECM remodeling during prostate development and carcinogenesis, (i.e., *Col1a1*, *Col1a2*^{2, 3, 4}), (b) epithelial-mesenchymal transition such as *Col4a1* and *Col4a2*^{5, 6} as well as (c) upregulated in the reactive stroma of CRPC (i.e., *Col6a2*, *Col6a3*)⁷. We added these concepts to the Results section (see **page 7**).

The idea that stromal cells reflect the AR activity of the nearby tumor cells (AR+ stroma in AR+

cancer models, AR- stroma in NEPC models) is quite interesting. It would be great to verify the abundance of POSTN+ vs AR+ stromal cells in additional GEM models of NEPC using IF/IHC.

Response: We agree with the Reviewer that this point is of great interest in dissecting the pathobiology of prostate cancer. Indeed, high *Postn* expression in clusters c5-c7 is inversely correlated with *Ar* expression in the stroma, and *Ar* expression is lowest in the PRN model. Immunofluorescence analysis confirmed high POSTN and low AR expression in the stroma of advanced GEMMs, especially around foci of neuroendocrine differentiation (see **Figure 5C-D**). In contrast, the highest expression of *Ar* and of its co-regulators (e.g., *Srebf1*) was found in c3 and c4, predominantly represented in T-ERG and Hi-MYC models. In response to the reviewer's comment, we have incorporated the same multiplex immunofluorescence (mIF) panel utilized in Figure 5 of our study (comprising chromogranin, AR, Periostin, and PanCK) to perform staining on tissues obtained from supplementary Genetically Engineered Mouse Models (GEMMs) of Neuroendocrine Prostate Cancer (NEPC). These supplementary models encompass PBCre4:*Pten*^{fl/fl}:*Rb1*^{fl/fl} (DKO) and PBCre4:*Pten*^{fl/fl}:*Rb1*^{fl/fl}:*Trp53*^{fl/fl} (TKO) strains⁸. Specifically, the DKO model, representative of adenocarcinoma prostate cancer, involves the targeted elimination of RB1, resulting in the induction of transformation to a variant that expresses neuroendocrine lineage markers. Furthermore, the introduction of an additional loss of TRP53 within this DKO framework leads to a complete transition of the disease into a fully antiandrogen-resistant neuroendocrine variant⁸. As seen in the PRN model previously, stromal cells in all models show high POSTN and low AR staining (see **Figure S4**) in areas adjacent to neuroendocrine differentiation. In addition to the mIF analysis, we also performed Visium spatial transcriptomics of prostate tissue from the PRN mouse model and its corresponding wildtype. Here, again, we found high *Postn* and low *Ar* expression in the stroma of PRN model compared to that of its WT (**Figure 5E**). Using Visium, we also found the expression of marker genes of PRN associated clusters (c5-c7) including *Mki67*, *Postn*, *Col12a1*, *Tnc*, and *Bgn* (see **Figure R2**).

Figure R2. Expression of marker genes for the PRN clusters (c5-c7) in the stroma of PRN mouse model compared to its WT.

The K-M plot in Figure 5G is impressive but I wonder how much of this is truly due to stromal cells in TCGA tissues used for sequencing and how much of it is due to the proliferative signature of these stromal cells. Can the authors computationally remove the proliferation genes and see if the more stromal-specific genes in the signature are driving the outcome (progression-free survival)?

Response: We thank the reviewer for raising this important point. Our signature is based entirely on genes derived from the stromal clusters derived mainly from the PRN mouse models (c5-c7) and does not include proliferative genes. Still the reviewer is right about the possible confounding by proliferative signatures. For this purpose, we tested a gene signature comprising genes involved in cell cycle proliferation, which were also shown to be associated with prostate cancer recurrence and mortality as shown by Cuzick et al.⁹. This signature consists of 31 cell cycle progression (CCP) genes and is reflective of the fraction of actively dividing cells with the tissue. We used this signature to predict distant metastases as well as to compute progression-free survival (PFS) in the TCGA cohort, using the exact same approaches that we adopted for the PRN gene signature. The PRN signature derived from the stroma of aggressive disease models is superior to the CCP signature at predicting PCa metastasis (**Figure S7A**). Furthermore, the CCP signature was significantly associated with PFS in univariate analysis (**Figure S7B**), but was not significant when adjusting for Gleason grade (**Figure S7C**). In contrast, the PRN signature was able to significantly capture PFS in both univariate and multivariate analysis (**Figure 5G**), which further underscores its robustness and translational value.

In Figure 6B, it appears as if most of these markers are expressed similarly across all stromal subtypes in contrast to data shown in mouse models in Figures 3A and 4A. Do the authors agree, and is there a hypothesis why this is the case?

Response: The Reviewer is right since the dot plot in Figure 6B (**Figure 7B** in the revised manuscript) was not reflecting the difference in gene expression levels between the 8 stromal clusters. In the revised version of the manuscript, we replaced this panel with violin plots showing this difference in expression levels together with statistical significance (p-value) resulting from a two-tailed t-test comparing the expression of these genes in their associated clusters versus the remaining clusters (**Figure 7B in the revised manuscript**). The violin plots in Figure 7B show a notable differential expression of marker genes across the different stromal clusters. For example, the highest expression of *ACTA2* and *MYL9* was in c0, while *FOS* and *JUN* had highest expression in c2, matching their expression patterns in the corresponding mouse data (**Figure 3A**). Similarly, *WNT4* and *RORB* were highly expressed in c3 and c4, while marker genes for the PRN-derived clusters including *POSTN* and *SFRP4* were highly expressed in c5-c7 (**Figure 7B**) which is in line with their expression in the mouse data (**Figure 4A**).

Several typos are noted. For example “of of” in the abstract and “stroma-epithelial interactions” on line 75 and “high frequency” on line 234. Please do a complete read-through for small edits and spelling before submitting the revised manuscript.

Response: We apologize for the errors. These have been corrected throughout the manuscript.

Reviewer #2:

Overall, I quite liked this manuscript. It is clear and well-written. The figures are nicely organized. The findings will help us understand the contribution of stromal cells in prostate cancer progression and metastasis and identify subpopulations of cells that may allow for therapeutic targeting.

The manuscript represents a valuable resource for the field. The manuscript is almost entirely descriptive and almost entirely based on transcriptomic data. This is not a criticism, but an observation. This could be published as a resource, and that resource would be useful for the research community.

Response: We thank the Reviewer for this positive assessment of our manuscript and its importance for the field. We agree that the manuscript can be perceived as descriptive since we highlight previously unknown stromal cell states and their role in mediating PCa initiation and progression. As can be seen in the response to Reviewer 1, we added functional experiments to substantiate some of the descriptive findings. Specifically, the experiments in vitro on Periostin demonstrate the involvement of this stromal gene in modulating the invasive potential of advanced/neuroendocrine carcinoma. In addition, we added a functional experiment to assess whether in co-culture experiments, epithelial cells with specific mutations determine changes in fibroblasts that recapitulate in gene expression place the clusters predominantly associated with these genotypes. Importantly, this is indeed the case (**Figure 5G**). We also examined the translational relevance of our findings in a cohort of PCa patients, where we used the transcriptional profiles of certain stromal cell states associated with advanced disease (c5-c7) to predict future metastatic events. This specific analysis revealed that the transcriptional profiles of these previously uncharacterized stromal cells can be predictive of distant metastatic events and is also significantly associated with survival independent of Gleason grade (**Figure 6**). We believe that the Reviewer is right about the value of our findings to the PCa scientific community. For this reason, we significantly expanded the supplementary data to include detailed information covering all aspects of our findings including the ligand-receptor interactions (**Table S3**), the transcriptional profiles of the stromal cell types (**Table S4**), and the PRN-derived signature (**Table S5**).

It would elevate the importance of the findings if the authors were able to validate key findings, e.g., by spatial transcriptomics, multiplex immunohistochemistry. As it stands, the biologic relevance of the author's findings is interesting but unproven.

Response: We thank the Reviewer for this valuable suggestion. In the revised manuscript, we performed spatial profiling of prostate tissues from mouse models and human cases to further validate our key findings. Specifically, we performed mIF analysis of additional Genetically Engineered Mouse Models (GEMMs) of Neuroendocrine Prostate Cancer (NEPC) to further confirm our initial findings in the PRN mouse model (reported in **Figure 5C**). These models encompass PBCre4:Pten^{fl/fl}:Rb1^{fl/fl} (DKO) and PBCre4:Pten^{fl/fl}:Rb1^{fl/fl}:Trp53^{fl/fl} (TKO) strains⁸. Specifically, the DKO model, representative of prostate adenocarcinoma, involves the targeted elimination of RB1, resulting in transformation to a variant that expresses neuroendocrine lineage markers but is morphologically mostly adenocarcinoma. Furthermore, the introduction of an additional loss of TRP53 within this DKO framework leads to a complete transition of the disease into a fully androgen-resistant neuroendocrine variant⁸ (**Figure S4**). In addition, we performed Visium spatial profiling of prostate tissue from the PRN mouse model and its WT and confirmed the mutual exclusivity of *Ar* and *Postn* expression, with PRN stroma displaying high *Postn* and low *Ar* expression, in contrast to the stroma of its WT (**Figure 5E**).

The validation data in Figure S6 (periostin and androgen receptor expression) is an attempt at validation, but is not very convincing in the current presentation and resolution, and lacks other stromal and tumor markers to confirm the spatial location where periostin and androgen receptor are being expressed.

Response: We thank the reviewer for raising this point. PanCK was used as an epithelial marker together with the other markers of interest including DAPI, AR, Periostin, and Chromogranin. This information was added to the revised manuscript (**Methods section, page 48**). Additionally, we have revised the figure (now **Figure S4 in the revised manuscript**) and added representative images of the whole tissue (H&E and mIF) and areas of interest, showing specifically PanCK, AR and Periostin staining. Violin plots are now included comparing the staining of AR and Periostin in stromal cells which shows a negative correlation between the expression of these two markers (high Periostin and low AR expression) (**Figure S4**). Finally, we also performed spatial transcriptomics profiling of prostate tissue from the PRN model and its wildtype using the 10x Visium platform. In that analysis, we again observed similar findings to those seen in the mIF analysis, with stromal cells in PRN showing a high *Postn* and low *Ar* expression, in contrast to wildtype stroma showing a high *Ar* and low *Postn* expression (**Figure 5E**).

SPECIFIC COMMENTS

The analysis of cell-cell communications is a little superficial. Besides the general signaling pathways, did the authors observe specific ligand and receptor interaction channels across the different stromal cell subsets? These specific channels could be listed. Are the cell communications pathways different between the three mouse models. Does the human data show similar interaction patterns?

Response: The reviewer is right that the cell-cell communications analysis was not detailed enough. We revised this analysis to show the individual ligand-receptor (L-R) interactions between the different cell types. In this revised analysis, we provided a more granular depiction of the LR interactions from epithelial and immune cells to the common stromal clusters (c0-c2) (**Figure 3C-E**). We also compared these L-R interactions in the GEMMs-specific clusters (c3-c7) in the corresponding mouse models and their wildtypes. We added a new supplementary table (**Table S3**) which includes all the significant L-R interactions in the TME of GEMMS. The L-R interactions analysis was limited to the mouse data only. Please see also response to Reviewer 1.

Epithelial cells are heterogenous, including different subsets such as luminal, basal, hillock and club. Is there a population of epithelial cells that **more closely interacted with stroma cells?** Would it be useful to include tumor cells into the analysis?

Response: We agree with the reviewer that different subsets of epithelial cells might have different interactions with stromal cells. To address this, we revised the L-R interactions analysis to provide a more granular depiction of the signaling networks from different subsets of epithelial cells including luminal, basal, and neuroendocrine cells (**Figures 3C, 4C, and 4D, and Table S3**).

Results, line 123, “an additional cluster was removed as it had <5% of cells in WT mice”. This immediately sounds like a tumor specific mesenchymal cell population that would be of particular interest. Is this the case, or do the authors simply mean that there were too few cells for a meaningful statistical analysis?

Response: Thank you for this insightful comment. Unfortunately, the number of wild-type (WT) cells included in the analysis was insufficient for conducting a robust statistical analysis. This observation brings up an additional hypothesis that remains to be validated, namely do the mesenchymal cell types giving rise to these clusters pre-exist in low numbers in the wild type setting and expand when adjacent epithelial cells are transformed or are the clusters a result of phenotypic shifts of uncommitted cells responding to the cues from the epithelium? Unfortunately, this very important question cannot be answered with the data available but will require lineage tracing using specific markers.

Figure 1A and 1E. I do not like the term “mutant(s)”, when I believe you are referring to the different GEMMs. Can this be labeled GEMMs for example?

Response: We thank the Reviewer for this suggestion. We revised this figure and labelled them as GEMMs instead of mutants.

Figure 1B and 1C could be moved to supplemental if space was needed.

Response: We thank the Reviewer for this valuable suggestion. We have now moved Figures 1B and 1C to the Supplementary Material (**Figure S2A-B**). **Figure S2A** sheds light on the total UMI counts per cell and the number of genes expressed in the count matrix which is important for confirming the proper preprocessing and quality control of our data. **Figure S2B** illustrates the percentage of distinct stromal cell clusters across the different mouse models which provides a quantitative assessment of the stromal composition in GEMMs and their WT.

Results, line 141. Figure S2 and Table S3. I like the idea of the regulons and the differential gene expression, though perhaps the authors could add 1-2 sentences to expand upon (a) how do the authors define a regulon, and (b) what criteria / cutoff identifies a differentially expressed gene in one cluster as compared to the other clusters? The authors use the regulon data quite a lot in subsequent results, and so it would be worth the time to expand on the explanation.

Response: We thank the reviewer for raising this important point regarding the gene regulatory networks (GRNs) and the differential gene expression (DGE) analyses. We feel that the GRNs analysis is important and further validates the clusters we identified. The methodology utilized is explained in detail in the Methods section, (**page 41**). The Results section has also been modified to add a brief description of what a regulon is based on the SCENIC pipeline (<https://doi.org/10.1038/nmeth.4463>) (**see Results section, page 8**). This section now reads: “*To this end, we performed cis-regulatory network inference to identify potential regulons, consisting of a TF and its putative targets, driving either genotypes or clusters. First, modules of highly correlated genes were identified, then pruned to include only those for which a motif of a shared regulator could explain the correlations. Subsequently, we scored the activity of each regulon in each cell and identified a set of regulons with different activity in the eight mesenchymal clusters (Figure S2C)*”.

For the DGE analysis, we compared the gene expression profiles of each cluster versus all the remaining clusters using the MAST (Model-based Analysis of Single-cell Transcriptomics) approach. This approach identifies DEGs between two cell groups using a hurdle model tailored to scRNA-seq data¹⁰. The analysis was limited to genes having at least 0.25-fold difference

Figure R3. Assessment of cell viability and health of 22rv1 shRb1 NMYC cells in Periostin Knockdown-derived medium. (A) 22rv1 shRb1 NMYC cell growth assessment. Data shown as number of viable cells plotted as % relative to control (n=2 in sextuplicate, mean \pm SEM) No significant effect of Periostin knockdown in fibroblasts on the viability of 22rv1 shRb1 NMYC cells has been shown. (B) Cells representing four different conditions (Negative control, shCtrl, shPostn1/2) were stained with Calcein-Orange and DAPI and then analyzed by flow cytometry in technical triplicates. Viable cells were considered Calcein-Orange positive and DAPI negative (Quarter 3; Q3 on FACS plots). Total percentage of viable cells confirms results in (A)

between the two cell groups and those that are expressed in at least 10% of cells in either groups. The DEGs between each cluster and the remaining clusters were ranked based on statistical significance (adjusted p-value) and the average log2 fold change (Log2FC) as shown in **Table S4**. The

corresponding Results section (Page 8) now reads: “We then

identified differentially expressed genes (DEGs) in each cluster versus the remaining clusters (**Table S4**) using the MAST approach which employs a hurdle model identify DEGs between two cell groups”. The corresponding Methods section (Page 41) now reads: “Using the default parameters, the DE analysis was limited to genes which show on average at least 0.25-fold difference between the two cell groups, and only genes that are detected in at least 10% of cells in either groups.”

Results, line 250. “In order to assess whether Postn-positive stroma facilitates invasion, a characteristic of NEPC, a migration assay was utilized.”. This is an intriguing finding, but I think it is an over-interpretation to say that stromal motility facilitates tumor cell invasion. Given that the migration is affected by POSTN knock-down, it would be important to show that the cells remained viable and otherwise healthy, confirming that the decreased migration was not simply due to unhealthy cells.

Response: It is crucial to ensure that any observed changes in migration are not compromised by viability issues. To address this concern, we performed an experiment using 22rv1 shRb1 NMYC cells. These cells were seeded in sextuplicate at a density of 200,000 cells per well in a 6-well plate and incubated in culture media for 24 hours. Subsequently, the cells were exposed to conditional medium derived from NAF (shCtrl, shPostn1, or shPostn2) cells for 6 hours. Cell viability was assessed by the Trypan Blue Dye Exclusion method. In addition, we performed Fluorescence-Activated Cell Sorting (FACS) analysis on the cells using Calcein-Orange which specifically labels live cells. DAPI was used to stain dead or damaged cells. Importantly, no significant effect of Periostin knockdown in fibroblasts on the viability of 22rv1 shRb1 NMYC cells was seen (**Figure R3**). Although a minimal increase in cell growth was observed, it was not

statistically. This finding was further confirmed by FACS analysis utilizing both calcein-orange and DAPI staining.

Results, line 262. “The transcriptional profiles of the PRN-derived clusters are predictive of metastatic progression in prostate cancer”. Here the authors took gene pairs expressed in a subset of the ~9000 mesenchymal cells (only clusters c5-c7). Then they used this signature to probe large bulk-RNA datasets, finding quite a nice AUC and quite a striking PFS difference. I worry about the significance of this finding, given that c5-c7 cells represent a maximum of 5% of the cells in a tumor (i.e., ~5000 cells out of ~100K cells that the authors sequenced) so it makes me wonder whether it is the expression within the mesenchymal cells driving these differences. Are those gene pairs expressed in other cells (i.e., tumor cells, immune cells, epithelial cells, etc.?) and is that possibly driving the findings or are these genes truly specific to the mesenchymal clusters c5-c7 or are they seen in other cell types within the tumor? The authors do have the data on the other 95,000 cells.

Response: We thank the Reviewer for raising this point. We hypothesized that primary tumor samples from patients who developed distant metastasis will have a more pronounced expression of the marker genes of the PRN-specific mesenchymal cells (c5-c7) compared to tumor samples from metastasis-free patients. The transcriptional profiles of the PRN-specific clusters (c5-c7) is specific to these stromal cells and was computed by comparing their expression profiles to other stromal cell types (see **Table S4** and **Methods section page 41**). Importantly, we built a list of gene pairs comprising both the top up-regulated ($\log_2FC > 0$) and down-regulated ($\log_2FC < 0$) genes in c5-c7. This list was used to constrain the training of the k-top scoring pairs (k-TSPs) classifier by limiting the feature selection to these gene pairs to select the most parsimonious set of pairs that can predict metastasis (see **Methods section, pages 44-45**). The resulting signature consists of 13 gene pairs, with each consisting of a gene that is up-regulated

Figure R4. The association between the PRN signature and progression-free survival (PFS) in the TCGA cohort based on tumor purity.

and another that is down-regulated in c5-c7 (**Table S5**), and the prediction rules depend on the relative ordering between the two genes. Patients predicted to have metastasis have a high expression of *gene1* (highly expressed in c5-c7 compared to other cell types) and a low expression of *gene2* (highly expressed in all cell types compared to c5-c7).

We also examined the prognostic performance of our signature based on tumor purity. Specifically, we compared the association between the PRN signature and progression-free survival (PFS) in the TCGA samples with the highest tumor purity (lowest stromal content) (**Figure R4A**) versus samples with the lowest tumor purity (highest stromal content) (**Figure R4B**). We

found that the association with survival with most significant in the samples with highest stromal content (log-rank p-value<0.0001) (**Figure R4B**). Furthermore, adding tumor purity as a co-variable in the multivariate COX proportional hazards model increased the hazard ratio of the PRN signature from 3.6 (95%CI=1.2-11) to 15.1 (HR=5.5-41.3) (**Figure R4C**). Overall, these results show that the predictive and prognostic performance of the PRN signature is driven mainly by the expression of its genes in stromal cells.

Finally, stromal cells with these expression profiles are significantly represented in the human data from bone metastasis, comprising ~70% of the total stromal cells (**Figure 7C**) underscoring their relevance to metastatic disease.

Cell type specific marker gene expression is not clear for c0 to c8. Many markers are commonly expressed in multiple clusters. (Figure 3A, 4A, 6B).

In Figure 4A, it seems that Wnt pathway related genes also show expression in C0 and c1. The DotPlot is not ideal for demonstrating the differences, we would recommend a violin type of plot with statistics.

Response: We thank the reviewer for raising this important point. To further provide a more accurate depiction of the expression levels of different marker genes across the eight stromal clusters, we provided a table of the differential expression results for each cluster compared to the remaining clusters (**Table S4**). For each cluster, this table shows the log2 fold-change (Log2FC) of each marker gene (including those depicted in **Figures 3A** and **4A**), together with p-value (p_val), adjusted p-value (p_val_adj), percentage of cells in the associated cluster expressing this gene (pct.1), and percentage of cells in the remaining clusters expressing this gene (pct.2). As outlined in the response to Reviewer 1 above, while there is a rank order of markers from highest to lowest expression that specify the different clusters (**Table S4**), the specificity of the top-ranked genes, is evident only in the GEMM-specific clusters (c3-c7).

While we agree with the Reviewer that dot plots may not be the best option for visualizing the difference in expression levels between different clusters, they offer the advantage of plotting a large number of genes displaying both their average expression (color bar) and percentage of cells expressing them (dot size), which is not feasible using violin plots. In addition to Table S4 providing comprehensive statistics about the marker genes expression, we also added a new supplementary figure (**Figure S5**) which includes violin plots of the expression of the top six marker genes for the common clusters, as well as the GEMMs specific clusters in the mouse and human scRNA-seq data.

It would be beneficial to create two gene expression plots which compare the mouse and human data projection with matched marker genes.

Response: Validation of the expression of marker genes in the human scRNA-seq data is indeed one of the main goals of this paper. A new supplementary figure (**Figure S5**) has been added. This Figure depicts the expression of the top marker genes for the common clusters (c0-c2), and the GEMMs-specific clusters (c3-c7) in both the mouse and human scRNA-seq data. This figure clearly shows the degree of conservation of the expression profiles of the mesenchymal clusters between murine models and the human mesenchyme when we utilized comparable genotypes, specifically Tmprss2-ERG.

Did you check NEPC single cell data to see if the three PRN/NEPC-(c5-c7) that were less abundant in human tumors are NEPC specific stromal cells? There exist public NEPC single cell datasets.

Response: We did not examine any public NEPC scRNA-seq data in our analysis. However, despite the low prevalence of the PRN associated clusters (c5-c7) in the human primary tumor data (**Figure 7A**), they were enriched in the human bone metastasis data (**Figure 7C**) and were also characterized by a high expression of *POSTN* and osteoblasts marker genes like *RUNX2*, *SPP1*, and *BGN* (**Figure 7D**). We hypothesized that these clusters, being derived mostly from the PRN mouse model, are associated with advanced disease, which explains their high abundance in the bone metastasis data (see **Results section, page 18**). Additionally, we examined the expression of *POSTN*, a canonical marker for these three clusters, in bulk RNA-seq data from Beltran et al. *POSTN* expression was highest in samples from patients with castration-resistant PCa (CRPC) and NEPC compared to benign and adenocarcinoma samples, further confirming our findings in the scRNA-seq data.

Did you observe any batch effect when combining bulk RNAseq datasets?

Response: Thank you for bringing up the important issue of batch effects when combining bulk RNAseq datasets. Indeed, we observed some batch effects initially and took a rigorous approach to correct these. Each dataset was individually normalized, log2-scaled, and z-score transformed before combining them into a large meta-dataset. This meta-dataset was then divided into training (75%) and testing (25%) cohorts using a stratified sampling approach. This approach ensured that both sets had adequate representation of important clinical and pathological variables, as well as the original datasets. Subsequent to this, we applied quantile normalization on the training and testing sets separately. We then evaluated the expression profiles of each of the training and testing sets using boxplots, which confirmed the successful removal of batch effects. We believe that these steps sufficiently mitigated the batch effects in our combined datasets and provided a reliable basis for our subsequent analyses. The steps detailing the preprocessing of the bulk RNAseq datasets including the removal of batch effect are described in the Methods section (page 44).

Statistical significance for Figure 6D is missing.

Response: We revised this Figure (now **Figure 7D**) to add statistical significance (p-values) resulting from a two-tailed t-test comparing the expression of *POSTN* and osteoblasts marker genes (*BGN*, *RUNX2*, and *SPP1*) in the stromal cell clusters associated with advanced disease (c5-c7) with their expression in all the remaining clusters in the bone metastases data from Kfoury et al.

Although the selected each GEMMs represent different pathological changes and disease stages, it would be great if the authors provided detailed information about the timing of harvesting the samples.

Response: In the presented studies, the comprehensive analysis of various pathological changes and stages of prostate cancer (PCa) necessitates meticulous consideration of multiple biological

Mouse model	Designation	time point of harvesting tissue
Tmprss2tm1.1(ERG)Sho	T-ERG (MT)	6 months
Nkx3.1creERT2;Pten ^{ff} ; EYFP ^{f/+}	NP (MT)	8 months
Tg(Arr2/Pbsn-MYC)7Key	Hi-MYC (MT)	6 months
Pb-Cre4 +/-;Pten ^{ff} ; LSL-MYCN +/-; Rb1 ^{ff}	PRN (MT)	8 weeks
FVB/N- pure background	WT for T-ERG and Hi-Myc	6 months
B6129SF2/J - littermates	WT for NP	8 months
C57BL6/129x1/SvJ - littermates	WT for PRN	8 weeks
C57Bl/6Jn- pure background	B6 (additional WT)	6 months
B6129SF2/J- pure background	B6.129 (additional WT)	6 months

Table for the Reviewer. *Genetically engineered mouse models and corresponding wild types at the analytical time-points used in the present study.*

variables, including genetic background, age, and the potential impact on animal welfare. These variables are acknowledged as confounding factors in this type of analysis and warrant careful management. Consequently, we have made a deliberate decision to maintain consistent genetic backgrounds among wild-type (WT) counterparts of genetically engineered mouse models (GEMMs) and to harvest them at the same age (as indicated in the table below). This rigorous approach was specifically applied to T-ERG, Hi-MYC, and their respective counterparts/littermates. However, due to the tamoxifen-inducible *cre loxP* system of the prostatic promoter Nkx3.1 specific to the NP cohort, recombination of the Nkx3.1cre line was activated after the completion of puberty in NP males, which typically occurs at 2 months of age, allowing for the development of the expected phenotype within 6 months. Unfortunately, the limited lifespan of PRN mice precludes reaching the desired age of 6 months. The predetermined clinical humane endpoint based on the Body Scoring Chart (BSC) index in mice for PRN mice is set within the range of 8-14 weeks. Consequently, PRN mice, along with their WT counterparts, were humanely sacrificed at 8 weeks of age.

REFERENCE FOR THE REVISION NOTE

1. Jin S, *et al.* Inference and analysis of cell-cell communication using CellChat. *Nat Commun* **12**, 1088 (2021).
2. Wang H, *et al.* Antiandrogen treatment induces stromal cell reprogramming to promote castration resistance in prostate cancer. *Cancer Cell* **41**, 1345-1362.e1349 (2023).
3. Wei X, *et al.* Paracrine Wnt signaling is necessary for prostate epithelial proliferation. *Prostate* **82**, 517-530 (2022).
4. Delliaux C, *et al.* TMPRSS2:ERG gene fusion expression regulates bone markers and enhances the osteoblastic phenotype of prostate cancer bone metastases. *Cancer Lett* **438**, 32-43 (2018).
5. Aytes A, *et al.* NSD2 is a conserved driver of metastatic prostate cancer progression. *Nat Commun* **9**, 5201 (2018).
6. Chan JM, *et al.* Lineage plasticity in prostate cancer depends on JAK/STAT inflammatory signaling. *Science* **377**, 1180-1191 (2022).
7. Zhu YP, Wan FN, Shen YJ, Wang HK, Zhang GM, Ye DW. Reactive stroma component COL6A1 is upregulated in castration-resistant prostate cancer and promotes tumor growth. *Oncotarget* **6**, 14488-14496 (2015).
8. Ku SY, *et al.* Rb1 and Trp53 cooperate to suppress prostate cancer lineage plasticity, metastasis, and antiandrogen resistance. *Science* **355**, 78-83 (2017).
9. Cuzick J, *et al.* Prognostic value of an RNA expression signature derived from cell cycle proliferation genes in patients with prostate cancer: a retrospective study. *Lancet Oncol* **12**, 245-255 (2011).
10. Finak G, *et al.* MAST: a flexible statistical framework for assessing transcriptional changes and characterizing heterogeneity in single-cell RNA sequencing data. *Genome Biol* **16**, 278 (2015).

REVIEWERS' COMMENTS

Reviewer #1 (Remarks to the Author):

The authors have responded to all of my comments and concerns with new experiments and/or descriptions and clarifications in the text and figures. I feel confident that the paper is much stronger now and will be easier for readers to understand. Overall, this work will serve as an outstanding resource for the field, and highlights important concepts to be studied further in the future about reciprocal signaling and regulation between mutated/transformed prostate epithelium and the tumor-adjacent stroma. I recommend accepting the manuscript for publication.

Reviewer #2 (Remarks to the Author):

Thank you for the nicely organized and very complete response to reviewer comments.

My concerns are addressed.

I know it can be hard given the space constraints, but most/all of the responses/explanations are great and really should be included in the manuscript (methods section? supplement) as they add detailed explanation, clarity, and a nice nuance to the manuscript.